# The actomyosin system is essential for the integrity of the endosomal system in bloodstream form *Trypanosoma brucei*

Fabian Link[1], Sisco Jung[1†], Xenia Malzer[1†], Felix Zierhut[2,3], Antonia Konle[1], Alyssa Borges[1], Christopher Batters[2,3], Monika Weiland[1], Mara Poellmann[1], An Binh Nguyen[1], Johannes Kullmann[1], Claudia Veigel[2,3], Markus Engstler[1*], Brooke Morriswood[1*]

[1]Department of Cell and Developmental Biology, Biocenter, University of Würzburg, Würzburg, Germany; [2]Ludwig-Maximilians-Universität München, Department of Cellular Physiology, Biomedical Centre (BMC), Planegg-Martinsried, Germany; [3]Center for Nanosciences (CeNS), München, Germany

*For correspondence:
markus.engstler@biozentrum.
uni-wuerzburg.de (ME);
brooke.morriswood@gmail.com
(BM)

†These authors contributed
equally to this work

Competing interest: The authors
declare that no competing
interests exist.

Reviewing Editor: Christine
Clayton, Centre for Molecular
Biology of Heidelberg University
(ZMBH), Germany

## eLife assessment

This **important** study builds on a previous publication, demonstrating that *T. brucei* has a continuous endomembrane system, which probably facilitates high rates of endocytosis. Using a range of cutting-edge approaches, the authors present **compelling** evidence that an actomyosin system, with the myosin TbMyo1 as an active molecular motor, is localized close to and can associate with the endosomal system in the bloodstream form of *T. brucei*. It shows **convincingly** that both actin and Myo I play a role in the organization and integrity of the endosomal system: both RNAi-mediated depletion of Myo1, and treatment of the cells with latrunculin A resulted in endomembrane disruption. This work should be of interest to cell biologists and microbiologists working on the cytoskeleton, and unicellular eukaryotes.

**Abstract** The actin cytoskeleton is a ubiquitous feature of eukaryotic cells, yet its complexity varies across different taxa. In the parasitic protist *Trypanosoma brucei*, a rudimentary actomyosin system consisting of one actin gene and two myosin genes has been retained despite significant investment in the microtubule cytoskeleton. The functions of this highly simplified actomyosin system remain unclear, but appear to centre on the endomembrane system. Here, advanced light and electron microscopy imaging techniques, together with biochemical and biophysical assays, were used to explore the relationship between the actomyosin and endomembrane systems. The class I myosin (TbMyo1) had a large cytosolic pool and its ability to translocate actin filaments in vitro was shown here for the first time. TbMyo1 exhibited strong association with the endosomal system and was additionally found on glycosomes. At the endosomal membranes, TbMyo1 colocalised with markers for early and late endosomes (TbRab5A and TbRab7, respectively), but not with the marker associated with recycling endosomes (TbRab11). Actin and myosin were simultaneously visualised for the first time in trypanosomes using an anti-actin chromobody. Disruption of the actomyosin system using the actin-depolymerising drug latrunculin A resulted in a delocalisation of both the actin chromobody signal and an endosomal marker, and was accompanied by a specific loss of endosomal structure. This suggests that the actomyosin system is required for maintaining endosomal integrity in *T. brucei*.

## Introduction

The actin cytoskeleton is a universal feature of eukaryotic cells and plays critical roles in processes such as endocytosis, cytokinesis, and maintenance of cell morphology. Most research has however focused on opisthokonts (principally animals and fungi), and very little is known about the actomyosin system in other branches of the eukaryotic tree, in particular the early branching ones. Recently, it was shown that trypanosomatids – a family of unicellular parasites – are the most divergent eukaryotes that still retain an actomyosin system with associated regulatory proteins (*Kotila et al., 2022*). Amongst the trypanosomatids, *Trypanosoma brucei* is the most tractable for molecular cell biology investigation. *T. brucei* is an exclusively extracellular pathogen which has evolved mechanisms to resist the harsh conditions within the bloodstream and body fluids of its mammalian hosts and the different but similarly demanding environments within its insect vector, the tsetse fly.

To evade the mammalian host's immune system, *T. brucei* bloodstream forms (BSFs) express a surface coat of variable surface glycoproteins (VSGs) consisting of $5 \times 10^6$ VSG dimers (*Bartossek et al., 2017*; *Cross, 1975*; *Jackson et al., 1985*). This dense glycoprotein coat hides invariant surface proteins from immune recognition, ensuring that host antibodies are generated against the currently presented VSG isoform and not against any invariant surface proteins (*Cross, 1975*; *Mugnier et al., 2016*). A glycosylphosphatidylinositol (GPI) anchor attached to each VSG monomer confers high mobility to the VSG coat, facilitating a high rate of antibody clearance by the parasite (*Bartossek et al., 2017*; *Borges et al., 2021*; *Hartel et al., 2016*). The directional movement of the trypanosome generates hydrodynamic flow forces on the cell surface that drag VSG-antibody complexes to the posterior region of the cell, where they can be endocytosed, trafficked to the lysosome, and degraded (*Engstler et al., 2007*; *Link et al., 2021*; *Webster et al., 1990*). Thus, *T. brucei* persistence inside the mammalian host is intrinsically connected to its endocytic trafficking capacity.

All endo- and exocytic processes in *T. brucei* occur through the flagellar pocket, a small invagination of the plasma membrane located near the posterior pole of the cell (*Grünfelder et al., 2002*; *Grünfelder et al., 2003*; *Lacomble et al., 2009*). Measurements of the kinetics of VSG endocytosis and recycling revealed that the VSG cell surface pool is turned over within 12 min (*Engstler et al., 2004*). Surface-bound antibodies are internalised and colocalise with lysosome markers within 2 min (*Engstler et al., 2007*). These rates are very high, especially considering that exo- and endocytosis are limited to only 5% of the total cell surface area (*Engstler et al., 2004*). Moreover, not only is the flagellar pocket situated in the posterior part of the cell, but the entire endosomal apparatus responsible for membrane recycling is also localised to that region (*Engstler et al., 2004*; *Field and Carrington, 2009*; *Grünfelder et al., 2003*; *Link et al., 2021*).

It was recently shown that the endosomal system of *T. brucei* has a remarkably complex architecture (*Link et al., 2023*). Rather than being divided into separate compartments, the system forms an interconnected membrane network. Within this network, distinct endosomal functions are grouped into membrane subdomains that are enriched with well-known markers associated with early (TbRab5A), late (TbRab7), and recycling endosomes (TbRab11). While this structure might explain the efficient membrane recycling seen in trypanosomes, the mechanisms responsible for maintaining its stability are not yet understood. One candidate for stabilising and organising this membrane network is the actomyosin system.

The actomyosin system of *T. brucei* is very reduced, consisting of one actin gene and two myosin genes (*García-Salcedo et al., 2004*; *Spitznagel et al., 2010*). This stands in contrast to its elaborate and highly structured subpellicular microtubule cytoskeleton (*Robinson et al., 1995*). Although the exact functions of the reduced actomyosin system in *T. brucei* are not understood, the retention of a minimal system (along with regulatory proteins) suggests an essential function that the microtubule cytoskeleton cannot recapitulate. Previous work on *T. brucei* actin showed that its depletion in BSFs resulted in a rapid cell division arrest, loss of endocytic activity, and ultimately cell death (*García-Salcedo et al., 2004*). Interestingly, *T. brucei* filamentous actin has not been visualised by phalloidin binding. This suggests that filamentous actin in trypanosomes may be short and dynamic, as is the case in *Plasmodium* (*Schmitz et al., 2005*; *Schüler et al., 2005*), and was recently shown in the related trypanosomatid parasite *Leishmania* (*Kotila et al., 2022*).

To date, the only experimental characterisation of trypanosome myosins was reported by Spitznagel and coworkers in 2010 (*Spitznagel et al., 2010*). That study focused on Tb927.4.3380 (TbMyo1), which belongs to the ubiquitous class I myosin family. Class I myosins are smaller than conventional

(class II) myosins, and do not self-associate into dimers or bipolar filaments. The functions of class I myosins vary from the regulation of membrane tension to participation in endo- and exocytosis, intracellular trafficking, and cell migration (reviewed in *Diaz-Valencia et al., 2022*). The myosin domain structure usually consists of a conserved motor domain with ATP- and actin-binding regions, a variable number of IQ (calmodulin-binding) motifs which activate the protein in a $Ca^{2+}$-dependent manner, and a cargo-binding tail domain (*Masters et al., 2017*; *Trivedi et al., 2020*).

In BSF *T. brucei* cells, TbMyo1 partially colocalised with endocytic cargo, late endosomes, and the lysosome (*Spitznagel et al., 2010*). In addition, the depletion of TbMyo1 resulted in the inhibition of cargo uptake and an enlargement of the flagellar pocket (*Spitznagel et al., 2010*). These findings supported a proposed function in endocytic trafficking. Although a direct colocalisation of actin and TbMyo1 was not demonstrated, the depletion or drug-induced disruption of actin resulted in a decreased and more diffuse TbMyo1 signal. It remains unclear, however, what kind of motor activity TbMyo1 has and whether it functions primarily as a tether or a transporter.

The other *T. brucei* myosin (Tb927.11.16310; TbMyo21, sometimes also referred to as Myo13) is part of the kinetoplastida-specific class XXI family (*Foth et al., 2006*). In *Leishmania*, knockdown of the class XXI myosin was reported to be lethal, with defects in flagellar assembly and a loss of intracellular trafficking (*Katta et al., 2009*; *Katta et al., 2010*). This critical role for Myo21 in *Leishmania* has led to interest in it as a potential drug target (*Trivedi et al., 2020*). In contrast, there has been no published work on TbMyo21, making it unclear whether it might also be a target for therapeutic intervention in *T. brucei*.

Here, a combination of different imaging techniques as well as biochemical and biophysical approaches were used to characterise both myosins in BSF *T. brucei* and determine their relationship to the endomembrane system and their corresponding marker proteins. As part of this work, an anti-actin chromobody system was established for the first time in trypanosomatids, thereby enabling simultaneous visualisation of actin and myosin. The results suggest that the actomyosin system of *T. brucei* is involved in post-endocytic membrane trafficking events and maintaining the complex membrane morphology of the endosomal system.

## Results

### TbMyo1 is distributed across cytosolic, organelle- and cytoskeleton-associated cellular fractions

Characterisation of TbMyo1 and TbMyo21 was initially carried out in parallel. To detect endogenous TbMyo1, two anti-TbMyo1 antibodies were generated against the tail domain (amino acids 729–1168) of the protein. The specificity of these antibodies was validated using two transgenic cell lines (*Figure 1—figure supplement 1*). The two anti-TbMyo1 antibodies were subsequently used as a 2:1 mix.

The same characterisation strategy was followed for the class XXI myosin. TbMyo21 was found to have extremely low expression levels in both BSF and procyclic life cycle stages, however. In addition, the protein could not be confidently localised using widefield microscopy and RNAi experiments showed that it appeared to be non-essential for BSF cell growth (*Figure 1—figure supplement 2*). Thus, all further experiments were conducted on TbMyo1.

To determine what proportion of TbMyo1 was associated with the cytoskeleton, a two-step biochemical fractionation was carried out (*Figure 1A*). Cells expressing cytosolic green fluorescent protein (GFP) were sequentially fractionated using the non-ionic detergents digitonin and IGEPAL. Digitonin preferentially solubilises the plasma membrane and leads to the release of the cytosolic fraction (SN1), while IGEPAL solubilises the organelle fraction (SN2). Samples were subsequently analysed by immunoblotting with fraction-specific marker proteins. As expected, the cytosolic GFP (~27 kDa) was efficiently extracted by digitonin and partitioned into the SN1 fraction (*Figure 1B*, top panel). The endoplasmic reticulum (ER) luminal chaperone BiP (~80 kDa) was not extracted by digitonin and mostly partitioned into the P1 fraction, but was subsequently solubilised by IGEPAL and partitioned into the SN2 fraction (*Figure 1B*, middle panel). The cytoskeleton-associated paraflagellar rod proteins TbPFR1 and TbPFR2 (~69 and 72 kDa) were not solubilised by either detergent and partitioned into first the P1 and then subsequently into the P2 fraction (*Figure 1B*, bottom panel, magenta). Interestingly, around half of the TbMyo1 (~130 kDa) partitioned into the SN1 fraction,

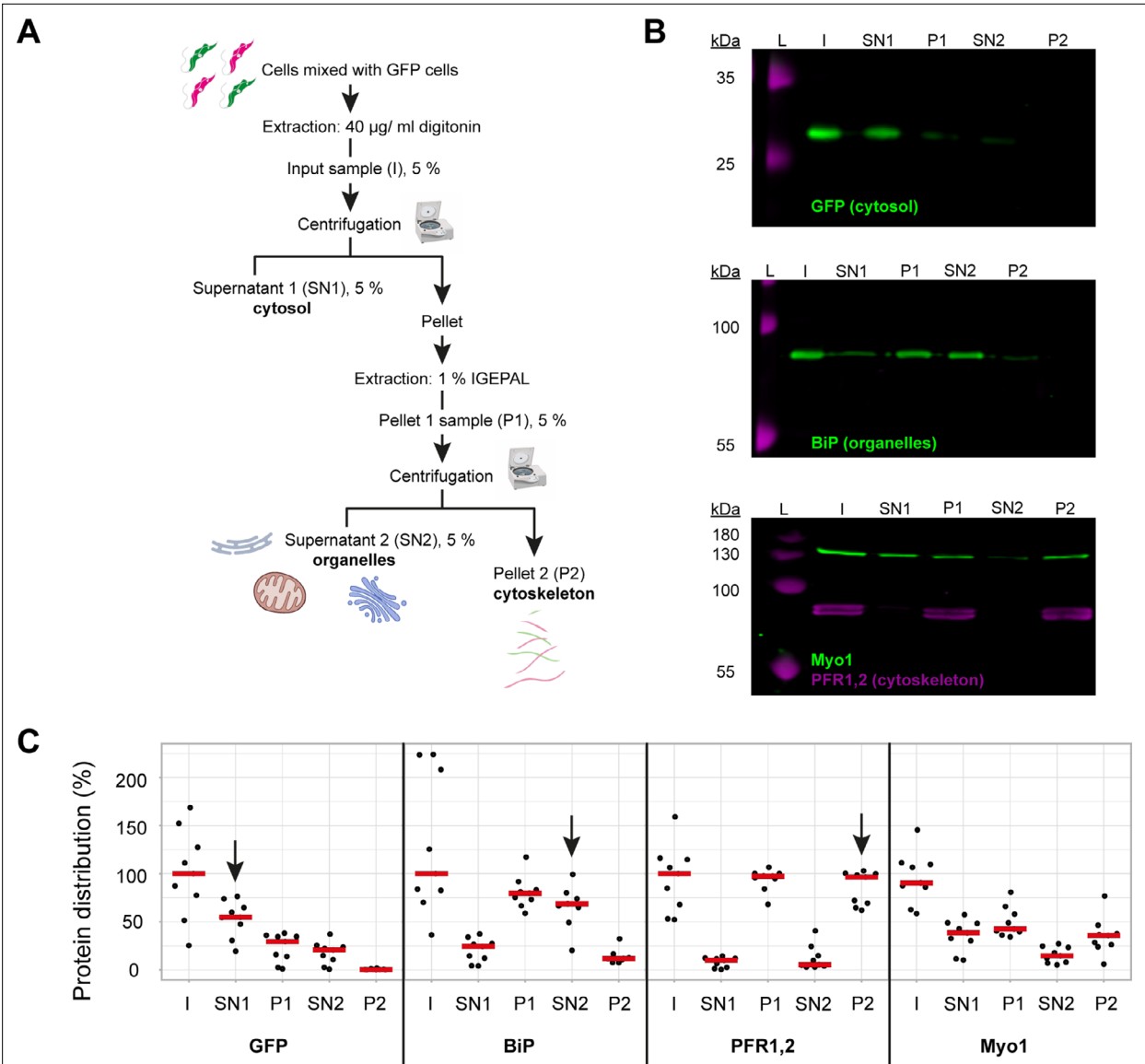

**Figure 1.** TbMyo1 has a large cytosolic pool. (**A**) Schematic representation of the two-step fractionation protocol. Bloodstream form (BSF) *T. brucei* cells were mixed with GFP-expressing BSF cells and extracted with digitonin to release the cytosolic fraction. The permeabilised cells were subsequently extracted with IGEPAL (non-ionic detergent) to release the organelle fraction. 5% samples of the input (I), supernatant (SN1, SN2) and pellet (P1, P2) fractions were taken at the indicated points. The samples were separated by SDS-PAGE and analysed by immunoblotting. (**B**) TbMyo1 is present in both a cytosolic and a cytoskeleton-associated pool. Exemplary immunoblotting results from the two-step fractionation protocol as described in (**A**). Equal fractions (~2%) were loaded in each lane. The cytosolic marker (GFP, ~27 kDa) was mainly detected in the SN1 fraction. The ER chaperone BiP (~80 kDa) partitioned into the P1 and then into the SN2 fractions. The flagellar cytoskeleton proteins PFR1,2 (~79 + 82 kDa) partitioned into the P1 and then the P2 fractions. TbMyo1 (~130 kDa) was found to be nearly equally divided between the SN1 and P1 fractions. The P1 fraction almost completely partitioned into the P2 fraction. Exemplary images from multiple (n = 8) independent experiments are shown. L, molecular weight ladder. (**C**) Nearly 50% of TbMyo1 is cytosolic. Quantification of the fractionation immunoblot data (n = 8). The red bars indicate the median.

The online version of this article includes the following source data and figure supplement(s) for figure 1:

**Source data 1.** PDF file containing original immunoblots for *Figure 1B* indicating the relevant areas; spreadsheet with raw data for panel C.

**Source data 2.** Original files for panel images displayed in *Figure 1B*.

**Figure supplement 1.** Generation and validation of TbMyo1 antibodies and cell lines.

**Figure supplement 1—source data 1.** PDF file containing original gels and immunoblots for *Figure 1—figure supplement 1B, C, D*, indicating the relevant areas.

**Figure supplement 1—source data 2.** Original files for panel images displayed in *Figure 1—figure supplement 1B, C, D*.

*Figure 1 continued on next page*

*Figure 1 continued*

**Figure supplement 2.** TbMyo21 is expressed at an extremely low level.

**Figure supplement 2—source data 1.** PDF file containing original gels and immunoblots for *Figure 1—figure supplement 2B, C, D,E* indicating the relevant areas; spreadsheet with raw data for *Figure 1—figure supplement 2F*.

**Figure supplement 2—source data 2.** Original files for panel images displayed in *Figure 1—figure supplement 2B, C, D,E*.

indicating that there is a substantial cytosolic pool. The remaining material (P1) predominantly partitioned into the cytoskeleton-associated P2 fraction (*Figure 1B*, bottom panel, green).

Quantification of multiple independent experiments indicated that 60% of the GFP signal was in the cytosol (*Figure 1C*, SN1, arrow). About 70% of the BiP signal was in the organelle-associated fraction (SN2, arrow) and 95% of TbPFR1,2 were in the cytoskeleton-associated fraction (P2, arrow). Also, 40% of TbMyo1 was found in both the cytosolic (SN1) and the cytoskeleton (P2) fractions. Only 20% of TbMyo1 was found in the organelle-associated (SN2) fraction.

## TbMyo1 translocates filamentous actin at 130 nm/s in vitro

The presence of a large cytosolic fraction of the monomeric TbMyo1 implied that it might be quite dynamic in vivo. Each class of myosins has its own kinetic properties, and consequently can function as either an active motor or a more static tether in vivo. The kinetic properties of TbMyo1 have not been investigated to date, so it was unclear whether it moved quickly or slowly on actin. To measure the velocity at which TbMyo1 can translocate actin, full-length recombinant TbMyo1 was purified and in vitro actin filament gliding assays were performed (*Figure 2*). Based on sequence analysis, TbMyo1 is predicted to consist of a large motor domain (cyan), a single calmodulin-binding IQ motif (yellow) and a membrane-binding FYVE-domain (magenta) (*Figure 2A*). The full-length protein was expressed in insect cells and purified, with most of the TbMyo1 eluting as a single peak with associated calmodulin (*Figure 2B and C*). For the in vitro gliding assays, the purified TbMyo1 was biotinylated, immobilised via streptavidin onto a glass surface, and incubated with Alexa488-phalloidin-labelled filamentous actin at 2 mM ATP (*Figure 2D*). The translocation of the filamentous actin by the immobilised myosins was observed and characterised quantitatively (*Figure 2E*, 'Materials and methods'). The actin filaments were found to move smoothly with a mean gliding velocity of 130 nm/s (*Figure 2—video 1*, *Figure 2E*). This indicated that TbMyo1 can functionally interact with actin and can translocate it at comparatively high speed.

## TbMyo1 is associated with the endocytic and not the biosynthetic pathway

The ability of TbMyo1 to actively translocate actin suggested that it might be functioning as a motor in vivo. To narrow down which pathways and cargoes it might be associated with, its localisation was examined using light microscopy. Previous studies indicated that TbMyo1 is located in the posterior region of the cell and overlaps with parts of the endosomal system (*Spitznagel et al., 2010*). These findings had not been made using high-resolution microscopy, however, and a potential localisation to the biosynthetic membrane trafficking pathway remained unexplored. Therefore, the localisation of TbMyo1 was examined using expansion microscopy.

After expansion of wild-type *T. brucei* cells, endogenous TbMyo1 was mainly detected in the posterior region of the cell between the nucleus (N) and the mitochondrial genome (kinetoplast, K) (*Figure 3A*). The fluorescence signal appeared as variable numbers of foci and larger clusters. Although a signal was detected in close proximity to the kinetoplast in some cells, the majority of signals were situated midway between the kinetoplast and the nucleus. These observations were consistent with the localisation described using widefield microscopy (*Spitznagel et al., 2010*).

To determine whether the TbMyo1 foci might be associated with biosynthetic pathways, colabelling experiments were carried out using the ER chaperone BiP, the Golgi marker GRASP, and the lysosome marker p67. The fixed, labelled cells were imaged using both widefield and structured illumination microscopy (SIM). No substantial overlap was seen between TbMyo1 and BiP (*Figure 3BI,II*). Accordingly, the correlation between the two proteins in the population was weak ($\rho = 0.35$) (*Figure 3C*). Similarly, TbMyo1 and the Golgi marker GRASP displayed no overlap (*Figure 3BIII,IV*) and weak correlation ($\rho = 0.30$) (*Figure 3C*). TbMyo1 and the lysosome marker p67 were frequently observed

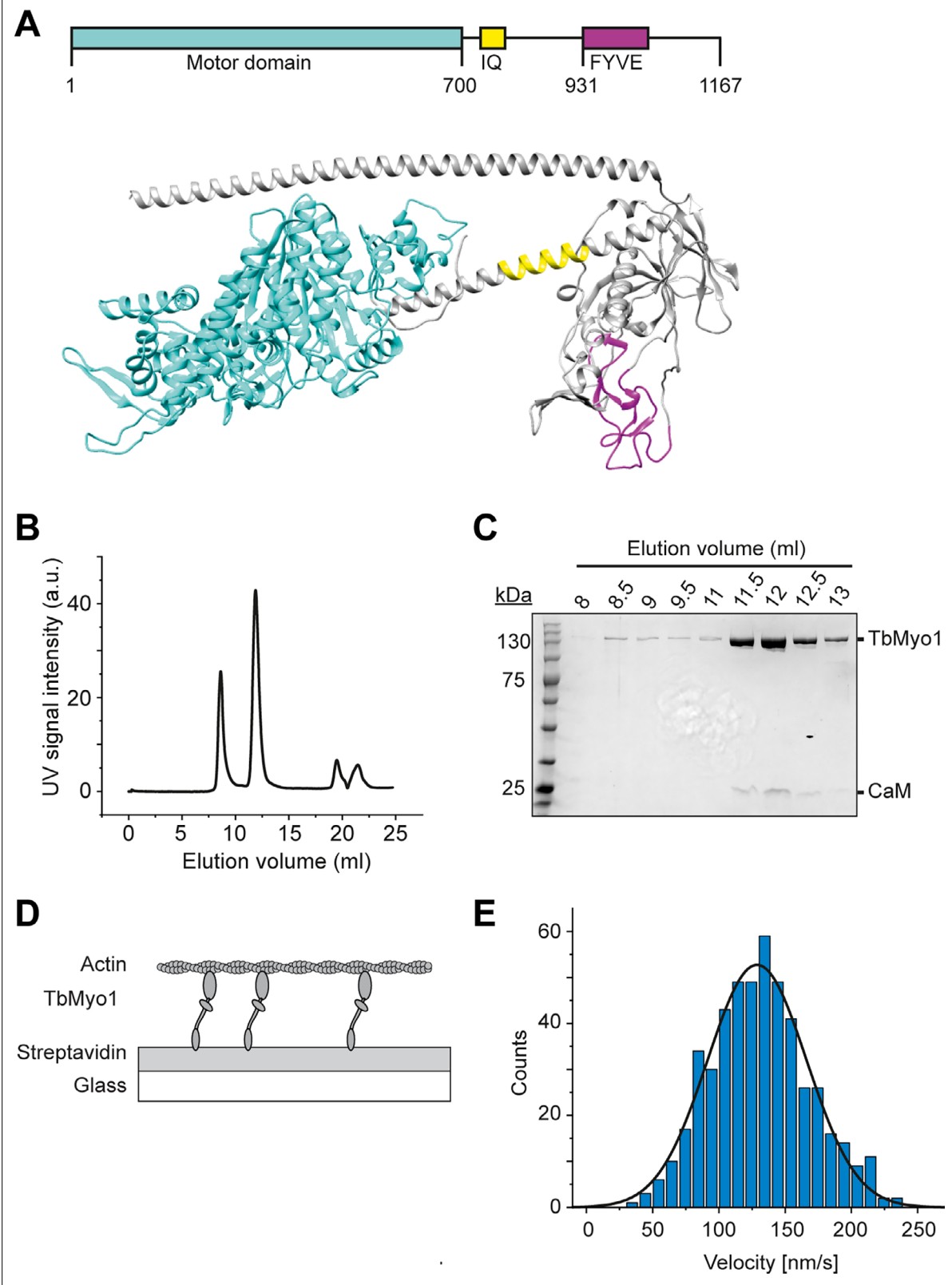

**Figure 2.** TbMyo1 translocates filamentous actin in vitro. (**A**) Schematic illustration and structural prediction of the domain architecture of TbMyo1. The motor domain is shown in cyan, the IQ (calmodulin-binding) motif is shown in yellow, and the FYVE domain is shown in magenta. Regions without domain prediction are displayed in grey. The structural prediction was generated using AlphaFold (ID: AF-Q585L2-F1). (**B**) Size-exclusion profile of affinity-purified TbMyo1 (Superdex 200 Increase 10/300). The monomeric TbMyo1 fraction eluted in a single peak at around 12 ml. The 9 ml (void

*Figure 2 continued on next page*

*Figure 2 continued*

volume) elution peak corresponds to aggregated protein. (**C**) TbMyo1 eluted mainly in the second protein peak. Samples were taken from the eluted volumes corresponding to the different protein peaks and analysed by SDS-PAGE. TbMyo1 (130 kDa) eluted together with human calmodulin in the 11.5–12.5 ml elution fractions. (**D**) Schematic representation of the in vitro actin filament gliding assay. Monomeric, C-terminally biotinylated TbMyo1 was immobilised via streptavidin on a glass surface. TbMyo1 translocated Alexa488-phalloidin labelled filamentous actin (F-actin) in the presence of ATP. (**E**) TbMyo1 translocates filamentous actin in vitro. Measurements were taken at 22°C with a final concentration of 2 mM ATP (n = 30 actin filaments). The average velocity was 130 ± 40 nm/s (mean ± SD; Gaussian fit).

The online version of this article includes the following video and source data for figure 2:

**Source data 1.** Spreadsheet with raw data for *Figure 2E*.

**Figure 2—video 1.** TbMyo1 translocate filamentous actin in vitro.

https://elifesciences.org/articles/96953/figures#fig2video1

in close proximity in the posterior region of the cell but did not overlap despite the TbMyo1 signal being close to and sometimes surrounding the p67 signal (*Figure 3BV,VI*). The correlation between the two proteins was moderate ($\rho$ =0.43) compared to the ER and Golgi markers and showed a much larger spread of values (*Figure 3C*).

To determine the distribution of TbMyo1 relative to endocytic cargo, fluorescent BSA and transferrin were used as markers (*Figure 3DI,II*). TbMyo1 had previously been suggested to overlap with the endosomal system, but this work was conducted using non-physiological cargoes (lectins) as markers (*Spitznagel et al., 2010*). Here, the physiological cargoes BSA and transferrin were used. The BSA signal was mainly found in a single globular cluster in the posterior region of the cell, which seemed to partially overlap with TbMyo1 (*Figure 3DI*). The transferrin signal was more widely distributed and seemed to have a more substantial overlap with TbMyo1 (*Figure 3DII*). Surprisingly, despite the clear visual overlap of cargo marker and TbMyo1 signals, the correlations were very weak ($\rho$ = 0.17 and 0.20, respectively) (*Figure 3E*). This might be because colocalisation can be measured in different ways, and correlation is not the same as co-occurrence (reviewed in *Aaron et al., 2018*). Correlation measures whether two distinct signals change in a similar manner within an overlapping area, while co-occurrence measures the extent of the overlapping area between the two signals. The results here suggest that although the TbMyo1 distribution overlaps with that of endocytic cargo, the signals are not strongly correlated. The implications are addressed in the 'Discussion.

In summary, TbMyo1 did not appear to associate with endomembrane organelles along the biosynthetic pathway (ER, Golgi, lysosome), but did overlap with endocytic cargo.

## Ultrastructural localisation of TbMyo1

The overlap between TbMyo1 and endocytic cargo strongly suggested that TbMyo1 is associated with the endosomal system, but there are other organelles present in the posterior region of trypanosomes. To test whether endogenous TbMyo1 is indeed primarily associated with the endosomal system, its ultrastructural localisation was investigated using cryo-sectioning and immuno-electron microscopy.

The morphology of the immuno-labelled structures was variable and included both vesicles and cisternae with circular or elongated shapes (*Figure 4A–D*). The vesicles were found in different sizes ranging from 30 to 100 nm, while the elongated membrane structures were between 20 and 30 nm thick and up to 1 µm long. Interestingly, glycosomes (readily identifiable because of their electron-dense lumen) were also regularly labelled with the anti-TbMyo1 antibodies (*Figure 4E*).

To test whether the membrane structures represented endosomes, the cells were pulsed with BSA conjugated to 5 nm gold as a marker for endocytic uptake (*Figure 4F–H*). The 5 nm gold particles were observed inside the membrane structures, confirming their endosomal identity (*Figure 4F–H*). In addition, gold particles were found inside coated vesicles and abundantly in the lysosome (*Figure 4H*).

The total number of gold particles associated with the different subcellular structures was quantified. While the majority of TbMyo1 gold signals were found in the cytoplasm (57%), equal proportions (15%) were found at glycosomes and endosomes, with smaller amounts at the flagellar pocket (8%) and on vesicles (5%) (*Figure 4I*). The large proportion of cytoplasmic TbMyo1 was consistent with the large cytosolic fraction previously observed in the biochemical fractionation assays (*Figure 1*).

To determine whether TbMyo1 was associated with a specific subdomain of the endosomal system, further colocalisation experiments were conducted using mNG-TbMyo1 tagged cells and antibodies

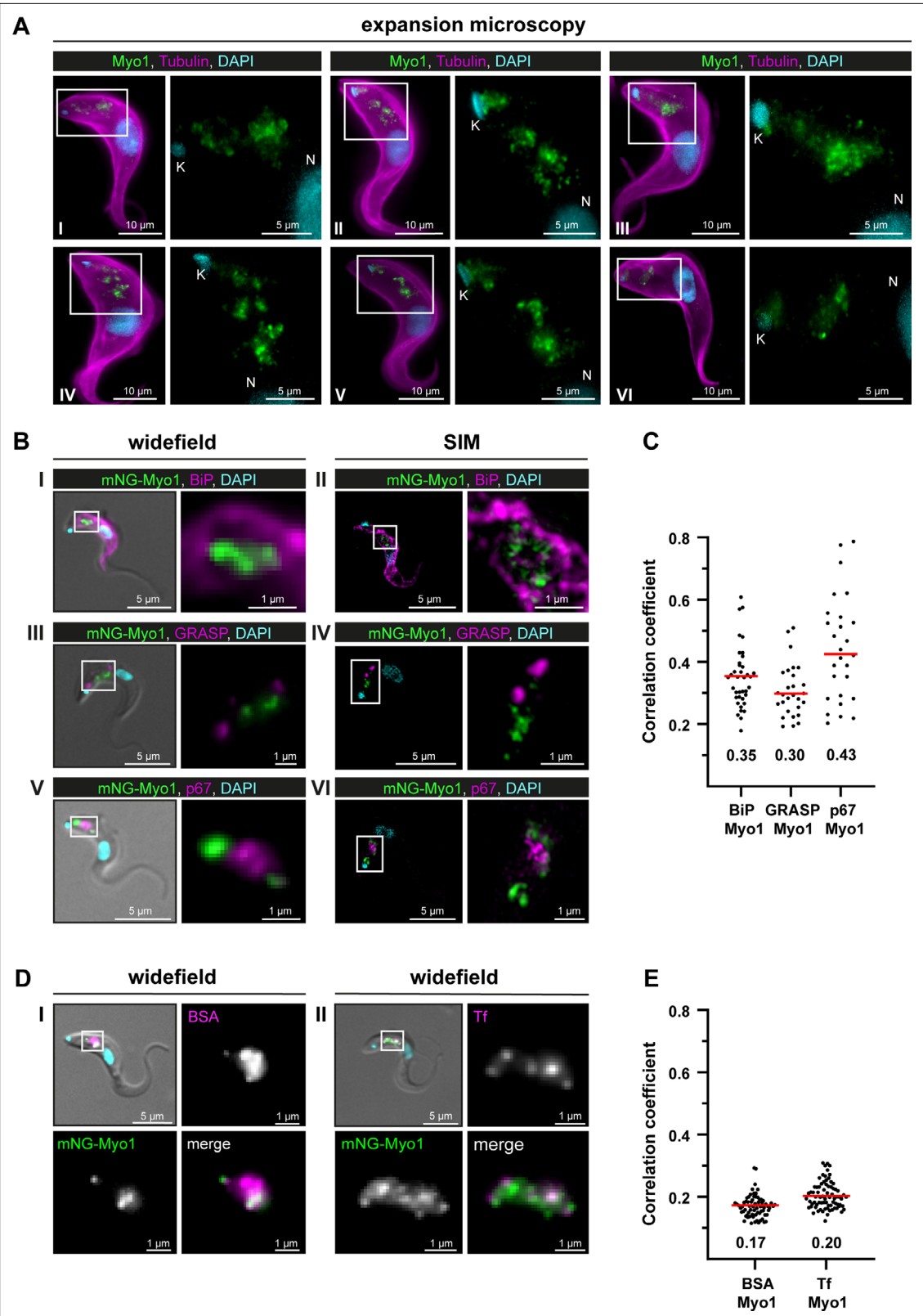

**Figure 3.** TbMyo1 is associated with the endocytic and not the biosynthetic pathway. (**A**) TbMyo1 is concentrated on structures in the posterior region of the cell. Expansion microscopy of bloodstream form (BSF) *T. brucei* labelled with anti-tubulin (yellow), anti-TbMyo1 (magenta), and DAPI (cyan). Six exemplary cells are shown (**I–VI**). For all cells, a magnified image of the posterior region (boxed area) is shown without the tubulin signal. The TbMyo1 signal was visible as a large cluster of foci in the posterior region between nucleus (N) and kinetoplast (K). (**B**) TbMyo1 does not colocalise with the ER

*Figure 3 continued on next page*

*Figure 3 continued*

(BiP, panels **I** and **II**), Golgi apparatus (GRASP, panels **III** and **IV**), or lysosome (p67, panels **V** and **VI**) markers. Fixed BSF cells were labelled using the indicated antibodies or tags and imaged using widefield microscopy and structured illumination microscopy (SIM). Overlay images with DIC and DAPI are shown for widefield microscopy images; SIM images show a merge of all three fluorescence channels. Magnified images of the regions of interest for TbMyo1 and the indicated markers are shown. (**C**) Quantification of correlation between TbMyo1 and organelle marker signals. Correlation was estimated using Spearman's rank correlation (BiP/TbMyo1 n = 38, GRASP/TbMyo1 n = 28, p67/TbMyo1 n = 27). The median $\rho$ is represented by a red line and the corresponding number is written below the data plot. (**D**) TbMyo1 overlaps with endocytic cargo. Cells expressing mNG-TbMyo1 were incubated with fluorescent cargo, fixed, and imaged using widefield microscopy. TbMyo1 (green) and the cargo markers (magenta) showed partial (BSA, panel **I**) or strong (transferrin, Tf, panel **II**) overlap. A magnified image of the fluorescence channels and a merge are shown next to the overlay with DIC. (**E**) Quantification of correlation between TbMyo1 and endocytic cargo marker signals. Correlation was estimated using Spearman's rank correlation (BSA/TbMyo1 n = 69, Tf/TbMyo1 n = 77). The median $\rho$ is represented by a red line and the corresponding number is written below the data plot. Exemplary images from multiple (n > 2) independent experiments are shown.

The online version of this article includes the following source data for figure 3:

**Source data 1.** Spreadsheets with raw data for *Figure 3C and E*.

against TbRab5A, 7, and 11 (*Figure 4—figure supplement 1*; *Link et al., 2023*). TbRab5A, 7, and 11 are markers of early, late, and recycling endosomal subdomains, respectively (*Hall et al., 2004*; *Hall et al., 2005*; *Silverman et al., 2011*; *Umaer et al., 2018*). TbRab5A and TbRab7 both partially overlapped with TbMyo1 (*Figure 4—figure supplement 1A*). The corresponding correlation coefficients exhibited large variation, underscoring the dynamic nature of both Rabs and TbMyo1 (*Figure 4—figure supplement 1B*). Immunogold labelling assays further demonstrated colocalisation between TbRab5A and TbRab7 with TbMyo1, highlighting that the proteins are found on the same membranous structures (*Figure 4—figure supplement 1C*). In contrast, the calculated correlation coefficient of 0.1 for the combination of TbMyo1 and TbRab11 reflected that these signals barely overlapped (*Figure 4—figure supplement 1A and B*).

In summary, ultrastructural analysis of TbMyo1 confirmed its association with early and late endosomal subdomains but not with the exocytic arms of the recycling compartment; TbMyo1 was additionally localised it to glycosomes.

## TbMyo1 is clustered adjacent to the lysosome

While the immunofluorescence data indicated no consistent overlap between TbMyo1 and the lysosome (*Figure 3B and C*), the immuno-electron microscopy data repeatedly showed TbMyo1 close to lysosomal membranes (*Figure 4H*, *Figure 4—figure supplement 1C*). To further explore this finding, correlative light and electron microscopy (CLEM) assays were used (*Figure 5*), as fluorophore-conjugated secondary antibodies can better penetrate cryosections than gold-conjugated ones (*van der Beek et al., 2022*).

To label the endosomes and lysosome, the cells were incubated with dual-labelled Alexa555-BSA gold prior to fixation and sample embedding. Anti-VSG antibodies (yellow) were used for cell surface labelling and DAPI (grey) to stain DNA, with both signals serving as fiducials for the correlation (*Figure 5I–III*). As the BSA gold was additionally conjugated to a fluorophore (Alexa555), its accumulation at the lysosome was first visualised using widefield microscopy (*Figure 5A and B*, cyan). The TbMyo1 signal (magenta) was frequently clustered on one side of the lysosomal signal but did not overlap with it. In addition, the TbMyo1 fluorescence was found to decorate endosomal membranes filled with gold particles, consistent with the earlier experiments (*Figure 5B*, arrow). Finally, TbMyo1 was again observed on glycosomes (*Figure 5C and D*, asterisks).

In summary, the use of CLEM on cryosections confirmed that TbMyo1 is clustered adjacent to the lysosome and is associated with endosomal membranes and glycosomes. This represents the first use of CLEM on cryosections in trypanosomes, confirming it as a viable experimental approach for *T. brucei*.

## Visualisation of TbMyo1 and actin overlap using an anti-actin chromobody

In previous studies, *T. brucei* actin was shown to localise to the endosomal system, and TbMyo1 localisation was shown to be dependent on actin (*García-Salcedo et al., 2004*; *Spitznagel et al., 2010*). Simultaneous imaging of the two proteins in *T. brucei* was not shown, however, and consequently, the

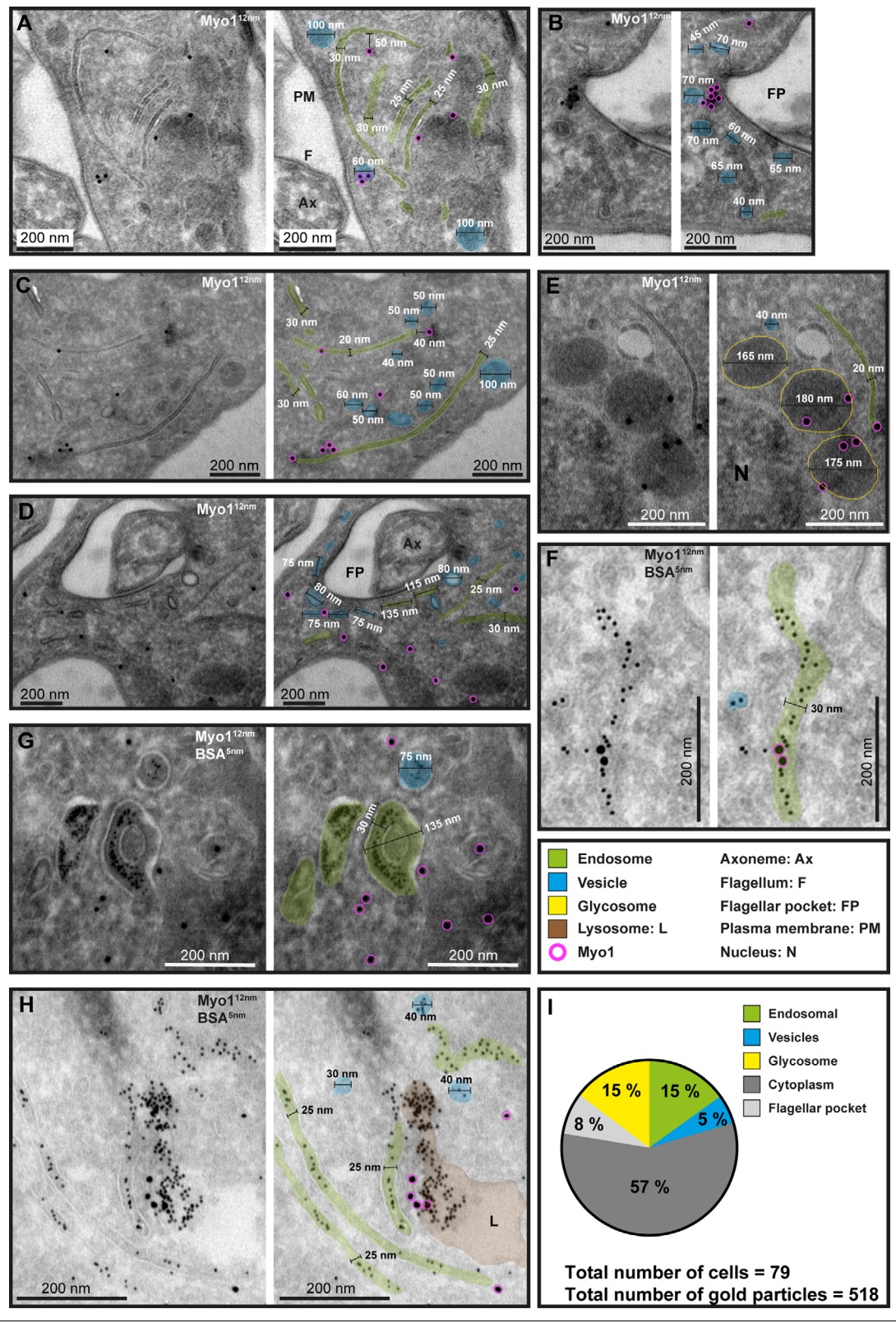

**Figure 4.** Ultrastructural localisation of TbMyo1. (**A–H**) TbMyo1 localises to endosomal tubes, vesicles, glycosomes, the flagellar pocket, and the cytoplasm. Electron micrographs of cryosections labelled with anti-TbMyo1 and 12 nm gold-conjugated secondary antibodies. For each panel, a raw and a pseudocoloured version of the image are presented. Endosomal tubes are coloured green. Vesicles are coloured blue. Glycosomes are outlined in yellow. The lysosome is coloured brown. Gold particles corresponding to TbMyo1 signals are outlined in magenta. (**F–H**) Cells were incubated with

*Figure 4 continued on next page*

*Figure 4 continued*

BSA conjugated to 5 nm gold particles prior to fixation and immunolabelling with anti-TbMyo1. The BSA was observed in vesicles, endosomes, and the lysosome. (**I**) Distribution of TbMyo1 in the cytoplasm and on subcellular structures. All gold particles 30 nm or closer to a structure were defined as being on the structure (based on a conservative calculation that two antibodies, each 15 nm in length, are between the bound epitope and the gold particle). More than half of the gold particles were found to be cytoplasmic. The total number of cells and gold particles quantified is indicated. Abbreviations: axoneme (Ax), flagellum (F), flagellar pocket (FP), plasma membrane (PM), and nucleus (N). Exemplary single- and double-labelled images were chosen from three separate labelling experiments.

The online version of this article includes the following source data and figure supplement(s) for figure 4:

**Source data 1.** Spreadsheet with raw data for *Figure 4I*.

**Figure supplement 1.** TbMyo1 is associated with early and late endosomes.

**Figure supplement 1—source data 1.** Spreadsheet with raw data for *Figure 4—figure supplement 1B*.

extent of colocalisation remained unknown. Given that fractionation data had indicated only 40% of TbMyo1 was cytoskeleton-associated, it seemed likely that the overlap between the two was not total.

Actin was previously visualised in *T. brucei* using an anti-actin polyclonal antibody (*García-Salcedo et al., 2004*). As an alternative approach, an anti-actin chromobody was tested here. Chromobodies are nanobodies (single-domain antibodies) conjugated to a fluorescent protein and can be expressed intracellularly. The anti-actin chromobody used here was previously used to visualise actin in living *Toxoplasma* cells (*Periz et al., 2017*). It had not been tested whether it could be used in trypanosomatids, however. If the anti-actin chromobody could be shown to work in trypanosomatids, this would potentially enable the study of dynamic actin localisations in live cells in the future.

The anti-actin chromobody was first subcloned into a trypanosome tetracycline-inducible expression vector and a stable cell line was generated. The correct insertion of the gene into the genome was validated using PCR (*Figure 6—figure supplement 1A*). Tight and inducible expression of the anti-actin chromobody at the population level was confirmed by immunoblotting (*Figure 6—figure supplement 1B*). At the single-cell level, a fluorescent chromobody signal was only detectable in the presence of tetracycline and was restricted to the posterior region of cells between the nucleus and kinetoplast (*Figure 6A*). This matched the previously reported localisation of actin in trypanosomes (*García-Salcedo et al., 2004*). Although there was cell-to-cell variation in the expression level of the chromobody, the distribution pattern was consistent (*Figure 4—figure supplement 1C*, *Figure 6B*).

To confirm that the chromobody signal was genuinely reporting on the distribution of filamentous actin, the cells were treated with latrunculin A (LatA) (*Figure 6C*). LatA binds actin monomers and prevents them from polymerising to filamentous actin, ultimately resulting in cell death. Previous work on TbMyo1 had already shown that it became delocalised when cells were treated with LatA (*Spitznagel et al., 2010*). While low concentrations (0.2 µM) of LatA showed a slight delocalisation of the chromobody signal, high concentrations (2 µM) resulted in a strong delocalisation or even loss of the chromobody signal. This strongly suggested that the chromobody was faithfully reporting on the distribution of filamentous actin in the trypanosome cells.

Next, the extent of overlap between TbMyo1 and actin was assessed. The signals strongly overlapped in the posterior part of the cell (*Figure 6D*). This was reflected in strong correlation between the signals in the population ($\rho$ = 0.63) (*Figure 6E*). Importantly, the TbMyo1 distribution was the same in uninduced and induced cells, indicating that its localisation was not affected by the expression of the chromobody. Of note, the colocalising area was often up to 1 µm away from the kinetoplast/flagellar pocket region (*Figure 6F*) and only a minority of cells showed a signal for either TbMyo1, actin, or both adjacent to the kinetoplast (*Figure 6FI,II*). These observations suggested that the actomyosin system is not constitutively present at the flagellar pocket.

## Actin maintains the morphology of the endosomal system

Lastly, the role of the actin cytoskeleton in stabilising the endosomal system was tested. Endosomes were labelled using ectopic expression of the marker protein EP1, which has previously been shown to label the entire endosomal system and flagellar pocket of BSF trypanosomes (*Engstler and Boshart, 2004*; *Link et al., 2023*). EP1, a cell surface protein found in insect-stage trypanosomes, is mostly excluded from the VSG coat of BSF trypanosomes.

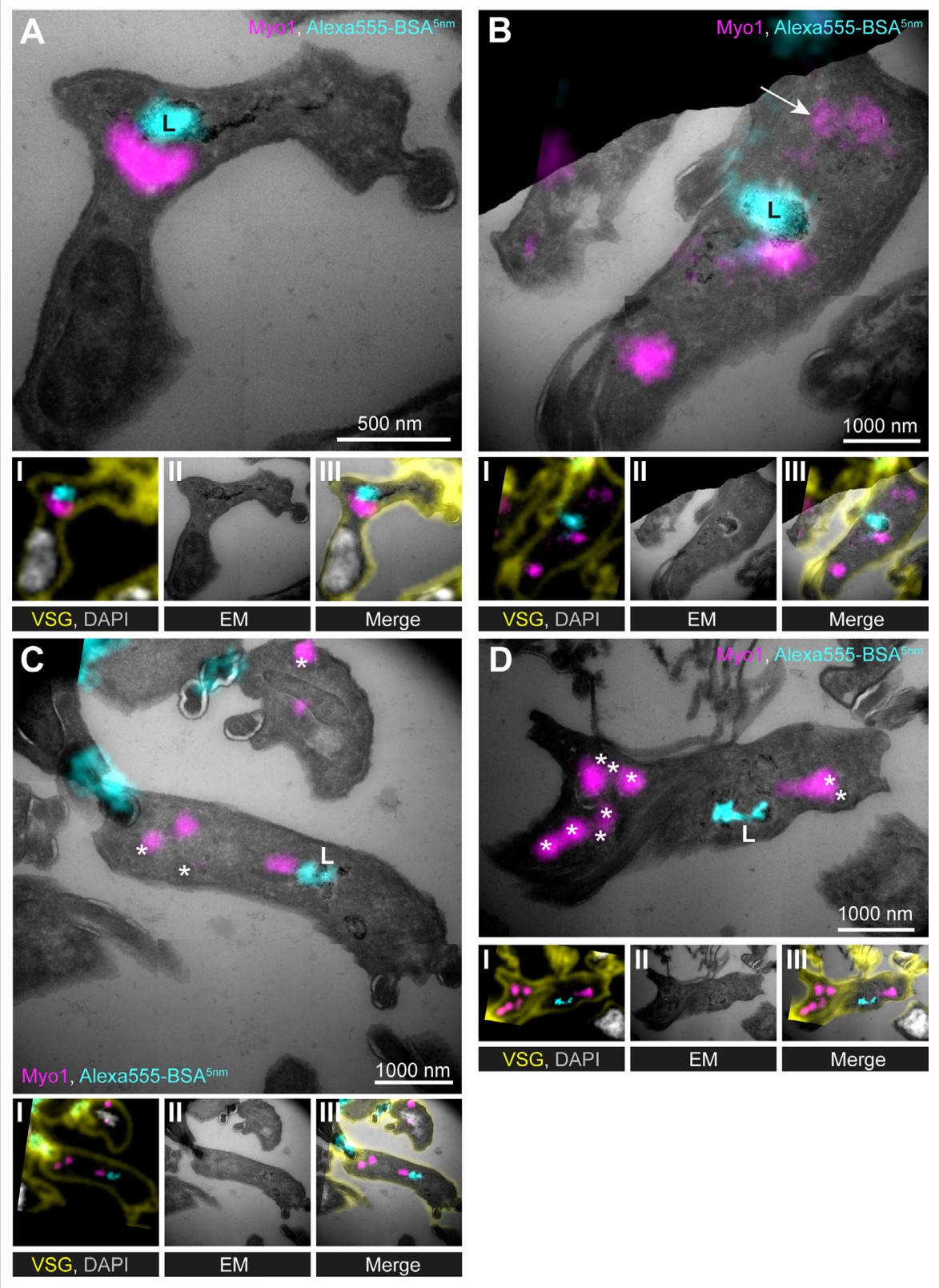

**Figure 5.** TbMyo1 is clustered adjacent to the lysosome. (**A–D**) Correlative light and electron microscopy (CLEM) imaging of cryosections. To visualise cargo trafficking, cells were incubated with double-labelled Alexa555- and 5 nm gold-conjugated BSA (cyan) before fixation. Cryosections were labelled with anti-TbMyo1 (magenta), anti-VSG (yellow), and DAPI (grey). For each panel, an image of the widefield microscopy (**I**), the electron micrograph (**II**) and an overlay of both images (**III**) is presented. The electron micrographs are manually stitched mosaics composed of multiple (4–9) tiled images of

*Figure 5 continued on next page*

*Figure 5 continued*

higher magnification (15,000–25,000×). Note that a fluorescence signal could only be detected at high local concentrations of Alexa555-BSA-5 nm gold. VSG and DAPI signals were used as fiducials to correlate IF and EM images. Correlations were made using the ec-CLEM plug-in in Icy for the initial correlation and Adobe Photoshop for the final overlay. Abbreviations: L, lysosome; *glycosomes; the arrow in (**B**) shows endosomal tubes filled with BSA gold and positive for TbMyo1. Images show exemplary images from a sample of 17 correlated cells.

To test whether the actomyosin system maintains endosomal morphology, the EP1-GFP-expressing cells were treated with 2 µM LatA for varying incubation times (*Figure 7*).

Control cells, harvested and fixed after 0 hr of LatA incubation, mostly (~85%) exhibited normal morphology. The EP1-GFP fluorescence in the controls was concentrated in the posterior region of the cell and revealed a variable number of high-intensity signals, surrounded and connected by weaker signals. In contrast, the effects of LatA incubation on the EP1-GFP signal mirrored its effects on the chromobody (*Figure 7A*). After 1 hr of LatA incubation, 50% of the cells displayed morphological changes, including swelling of the cell body in the posterior part and even complete rounding of the cell (*Figure 7B*). After a 2 hr incubation with LatA, only 10% of the cells exhibited a typical morphology. At longer incubation times, no cells displaying a normal morphology were observed. These morphological changes were accompanied by a redistribution of the EP1-GFP fluorescence signals. Instead of being confined to the posterior region, the EP1-GFP signals increasingly spread throughout the entire cytoplasm. Examination of the ultrastructure of the endosomal system at early timepoints (≤60 min LatA incubation) revealed a complete loss of endosomal integrity, resulting in large membrane vesicles filling the space between the flagellar pocket and the lysosome (*Figure 7C*). The preservation of the Golgi complex structure in its entirety, as well as the continuing invagination of clathrin-coated pits and presence of clathrin-coated vesicles at the flagellar pocket, underscored the specificity of this phenotype (*Figure 7C*). It is noteworthy that even after 30–60 min of LatA incubation, which led to the complete disruption of the endosomal system, the parasites remained motile. Moreover, an enlargement of the flagellar pocket was not observed in most cells during this period.

To test whether TbMyo1 also contributed to endosomal system morphology, experiments were carried out using the TbMyo1 RNAi cell line. Depletion of TbMyo1 caused a very penetrant phenotype, with most of the cells already rounded-up after 24 hr of induction (*Figure 7—figure supplement 1A*). These rounded-up cells are not phenotypically informative, but the cells with a less deranged morphology were still capable of internalising BSA and trafficking it to a lysosome-adjacent compartment (*Figure 7—figure supplement 1A*). Depletion of TbMyo1 was confirmed by immunoblotting (*Figure 7—figure supplement 1B*). Population growth curves matched previously published results and indicated that 24 hr post-induction was an appropriate starting point for phenotypic investigation (*Figure 7—figure supplement 1C* and *Spitznagel et al., 2010*). In a preliminary electron microscopy analysis of TbMyo1-depleted cells, enlarged endosomal structures were occasionally seen, consistent with the observations made using LatA-treated cells (*Figure 7—figure supplement 1D*). As in the LatA experiments and consistent with the BSA uptake experiments, clathrin-coated vesicles in the cytoplasm and clathrin patches on the flagellar pocket membrane were frequently observed, suggesting that endocytosis was not strongly affected (*Figure 7—figure supplement 1D*). The data therefore specifically suggest that the complex architecture of the endosomal system in trypanosomes is maintained by the actomyosin cytoskeleton.

## Discussion

Despite the considerable research focused on the microtubule cytoskeleton of *T. brucei*, its actomyosin system has received relatively little attention. The polarised distribution of actomyosin system proteins to the posterior part of *T. brucei* and the recent discovery that the endosomal system of this organism is a three-dimensional network rather than isolated compartments make the actomyosin system a plausible candidate for maintaining endosomal organisation (*Link et al., 2023*). In this study, the arrangement of myosins and actin in BSF *T. brucei* was revisited using a combination of imaging and biophysical techniques.

Initially, both the class I (TbMyo1) and trypanosomatid-specific class XXI (TbMyo21) proteins were investigated. TbMyo21 exhibited minimal expression in both BSF and procyclic form cells, however, and no detrimental growth effects were apparent upon depleting TbMyo21 in BSF cells (*Figure 1—figure*

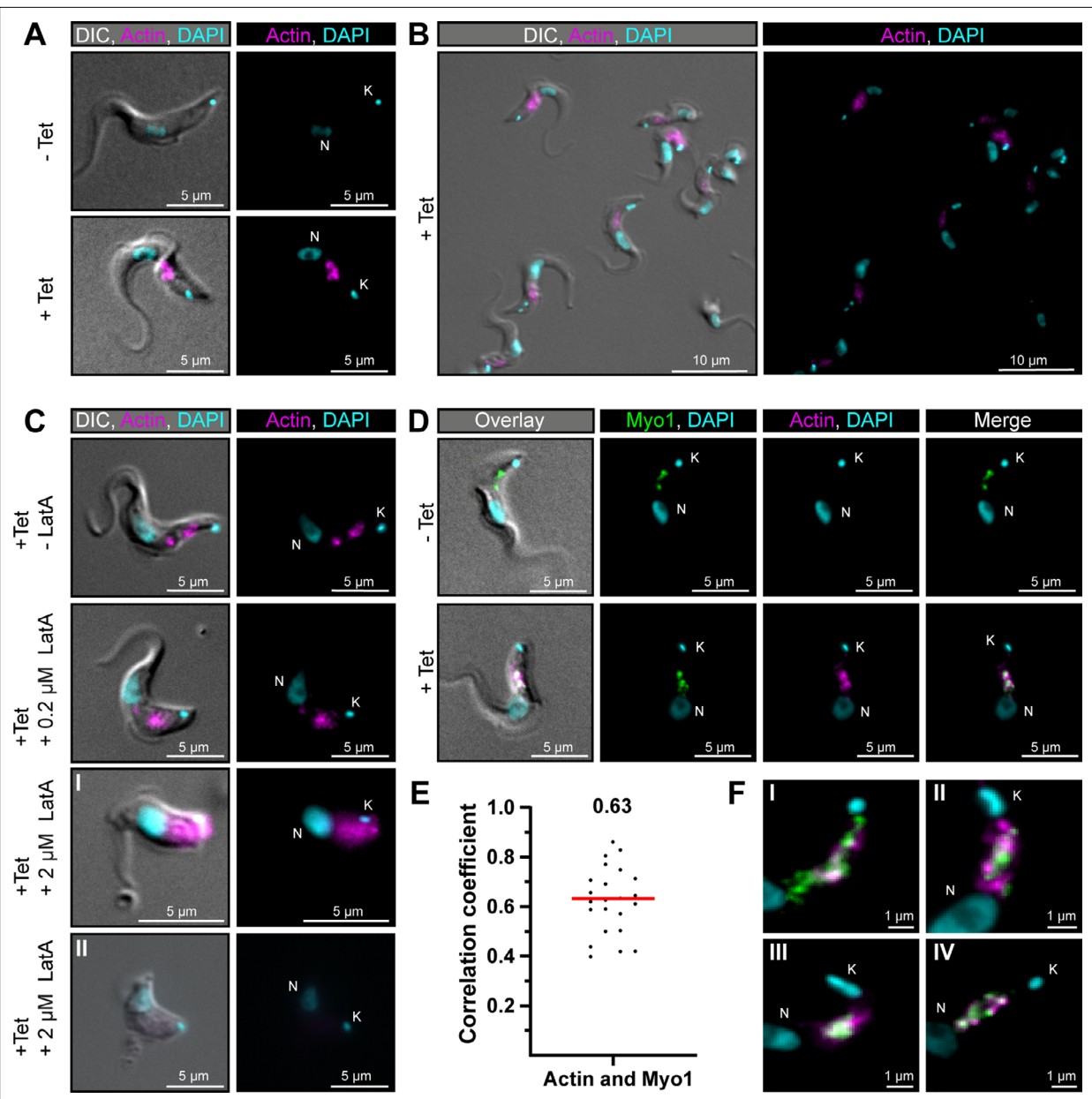

**Figure 6.** TbMyo1 overlaps extensively but not completely with actin. (**A**) Imaging of trypanosome actin using an inducibly expressed anti-actin chromobody. Tetracycline (Tet) was used to induce expression of the anti-actin chromobody. Widefield microscopy of fixed cells in the absence (- Tet) or presence (+Tet) of tetracycline are shown. The induced (+Tet) cells showed a chromobody signal (magenta) in the posterior region of the cell between the nucleus (N) and kinetoplast (K), consistent with the reported localisation of actin in BSF *T. brucei*. DNA was stained with DAPI. (**B**) Cell-to-cell variation of the chromobody signal in the tetracycline-induced population. The intensity but not the distribution of the signal varied from cell to cell. (**C**) The actin depolymerising drug latrunculin A (LatA) disrupts the chromobody signal. Widefield microscopy of fixed, induced chromobody cells in the presence of different LatA concentrations. The cells in the absence of LatA (- LatA) acted as controls. The addition of 0.2 µM LatA led to a slight blurring of the chromobody signal (magenta), while 2 µM LatA resulted in strong blurring (**I**) or even loss of signal (**II**) accompanied by a disruption of cell morphology. (**D**) TbMyo1 strongly overlaps with the anti-actin chromobody signal. Non-induced cells acted as a control and showed no chromobody signal, with only a TbMyo1 signal (green). Induced cells displayed both a chromobody (magenta) and a TbMyo1 signal, which strongly overlapped in the posterior region of the cell. (**E**) Quantification of correlation between TbMyo1 and actin signals. Correlation was estimated using Spearman's rank correlation (n = 23). The median $\rho$ is represented by a red line and the corresponding number is written above the data plot. (**F**) Enlarged view of the strong overlap between TbMyo1 and anti-actin chromobodies in the posterior region of the cell. Magnified images of the posterior region of four different cells (**I–IV**) are shown. TbMyo1 and the anti-actin chromobodies strongly overlapped in all cells examined, but the number and intensity of discrete spots varied.

The online version of this article includes the following source data and figure supplement(s) for figure 6:

*Figure 6 continued on next page*

*Figure 6 continued*
**Source data 1.** Spreadsheet with raw data for *Figure 6E*.
**Figure supplement 1.** Generation and validation of the anti-actin chromobody cell line.
**Figure supplement 1—source data 1.** PDF file containing original gels and immunoblot for *Figure 6—figure supplement 1A and B* indicating the display areas.
**Figure supplement 1—source data 2.** Original files for panel images displayed in *Figure 6—figure supplement 1A, B, and C*.

*supplement 2*), consistent with previously published high-throughput data (*Alsford et al., 2011*). It remains possible that TbMyo21 might be important only in cell cycle stages that cannot be cultivated in vitro, such as epimastigotes. Due to its exceedingly low expression levels and non-essential role in BSF *T. brucei*, TbMyo21 is unlikely to be a viable drug target, in contrast to its homologue in *Leishmania* (*Trivedi et al., 2020*). Subsequent efforts in this study focused on the characterisation of TbMyo1. Of note, preliminary data indicate downregulation of TbMyo1 expression in the procyclic life cycle stage (*Figure 1—figure supplement 2E*), consistent with earlier proteomic experiments (*Tinti and Ferguson, 2022*). This finding differs with the previous study, where no change in TbMyo1 expression levels was observed after differentiation (*Spitznagel et al., 2010*). The discrepancy is most likely due to the stronger signal obtained in the present study with the new anti-TbMyo1 antibodies, although a more rigorous experiment using a loading control with identical expression levels in the two life cycle stages is still needed. This nonetheless suggests that the actomyosin system is not as critical for the viability of procyclic cells, which have a much lower rate of membrane trafficking than the BSF life cycle stage.

The biochemical fractionation data indicated that around half of the TbMyo1 was in the cytosolic fraction, while the remaining portion exhibited a strong association with the cytoskeleton fraction (*Figure 1*). The substantial cytosolic fraction of TbMyo1 could potentially indicate the molecules adopting an auto-inhibited 'foldback' configuration (*Yang et al., 2009*). This is speculation, however, and needs to be tested in future in vitro assays. The existence of a large cytosolic fraction at steady state might also suggest that TbMyo1 is very dynamic, with a high turnover on the organelles it is associated with.

The in vitro motility assays indicated that TbMyo1 can translocate filamentous actin with a mean translocation velocity of 130 nm/s, which is significantly faster than the approximately 10–50 nm/s measured for some mammalian class I myosins (*Giese et al., 2021*; *Pyrpassopoulos et al., 2012*). This relatively high velocity might indicate that TbMyo1 is participating in intracellular trafficking of BSF *T. brucei* and functioning as an active motor rather than a static tether. It should be stressed, however, that this study has not investigated any of TbMyo1's biochemical or mechanical activities in vivo, so this remains a hypothesis for now. The in vitro motility assay conditions obviously do not replicate the highly complex and dynamic cytoplasmic environment within a trypanosome cell, and multiple regulatory mechanisms, binding partners, and maybe even membrane interactions (see below) will all come into play.

Earlier studies examining the localisation of TbMyo1 primarily concentrated on the endocytic pathway (*Spitznagel et al., 2010*), with little attention given to its potential association with the biosynthetic route. Nevertheless, neither widefield microscopy nor SIM displayed any notable overlap between TbMyo1 and markers of the biosynthetic pathway (*Figure 3B and C*). The close juxtaposition of endoplasmic reticulum exit sites (ERES) and the Golgi apparatus in *T. brucei* could provide a rationale for why a motor protein like TbMyo1 might be dispensable for mediating transport between these organelles (*Sealey-Cardona et al., 2014*). While myosins have often been localised to the Golgi in other systems (*Avisar et al., 2008*; *Buss et al., 1998*; *Perico et al., 2021*), these cell types typically have multiple Golgi apparatuses and a more intricate coordination of secretory pathways.

Recently, a potential mechanism was discussed in which secretory material in *T. brucei* might move in bulk from the Golgi to the endosomal system (*Link et al., 2023*). The secretory material would subsequently travel alongside recycled endocytic cargo from the endosomal system to the flagellar pocket and from there to the surface plasma membrane. In this model, it is the endosomal network that serves as a central sorting hub for both secretory and endocytic cargo. Maintenance of its complex three-dimensional morphology might therefore be critical to efficient function, and this is a process that the TbMyo1 and actin could plausibly contribute to.

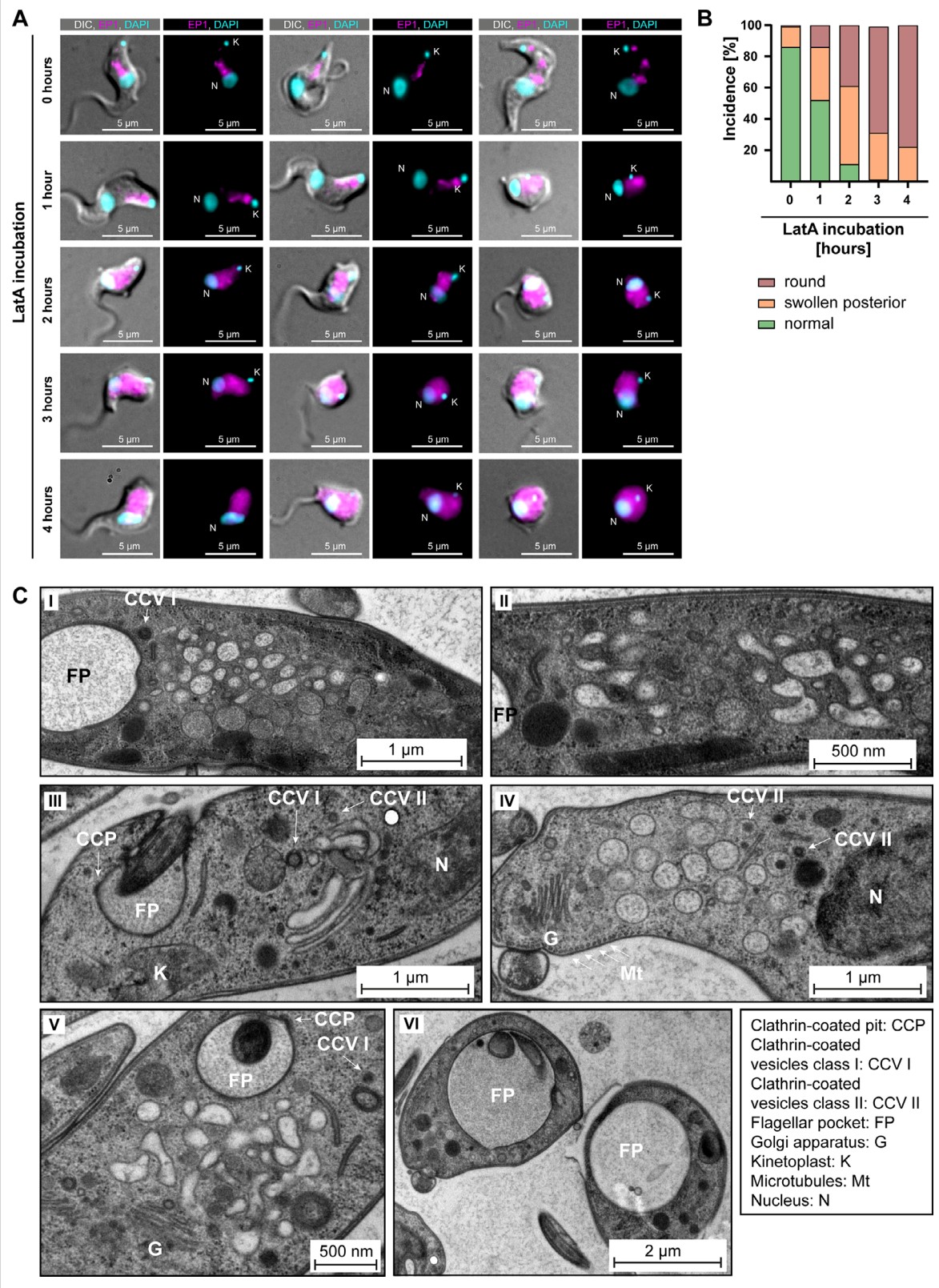

**Figure 7.** Actin depolymerisation disrupts the endosomal system. (**A**) Latrunculin A (LatA) treatment perturbs the endosomal system. EP1-GFP-expressing cells were incubated with 2 µM LatA for the indicated time periods, then fixed and imaged using widefield microscopy. DNA was stained using DAPI. Longer incubation with LatA resulted in aberrant cell morphologies and an increasingly diffuse EP1-GFP signal. Exemplary cells from a single experiment are shown. (**B**) Quantification of the morphological effects of LatA treatment. Cells were manually classified as displaying normal,

*Figure 7 continued on next page*

*Figure 7 continued*

swollen posterior, or rounded morphology. The degree of morphological disruption was proportional to the duration of incubation with LatA. Data acquired from >200 cells assayed in three independent experiments. (**C**) LatA treatment disrupts the ultrastructural morphology of the endosomal system. Cells expressing EP1-GFP were subjected to a 30–60 min incubation with 2 μM LatA, followed by high-pressure freezing, embedding in Epon, and examination through transmission electron microscopy. The cytoplasm contained many large and interconnected tubular profiles, consistent with a swollen endosomal system. An enlargement of the flagellar pocket was additionally observed. Exemplary cells from a single experiment are shown.

The online version of this article includes the following source data and figure supplement(s) for figure 7:

**Source data 1.** Spreadsheet with raw data for *Figure 7B*.

**Figure supplement 1.** TbMyo1 depletion affects cell morphology and endosomal membrane organisation.

**Figure supplement 1—source data 1.** PDF file containing original gels and immunoblots for *Figure 7—figure supplement 1B, C, D*, indicating the relevant areas.

**Figure supplement 1—source data 2.** Original files for panel images displayed in *Figure 7—figure supplement 1B, C, D*.

The actomyosin system has often been associated with membrane manipulation in different organisms (*Costa et al., 2020*; *Eitzen, 2003*; *Khatibzadeh et al., 2013*; *Khanal Lamichhane et al., 2019*). Interestingly, it has recently been shown that in *Drosophila* endosomes can organise F-actin and that endosomal actin can affect subcellular tube growth and integrity (*Ríos-Barrera and Leptin, 2022*). Thus, it could be that the actomyosin system either generates or helps maintain the architecture of the trypanosome endosomal system. Furthermore, the predicted FYVE domain in the TbMyo1 tail domain is predicted to mediate binding to PI(3)P lipids, which are enriched in endosomal membranes in other organisms (*Marat and Haucke, 2016*). Preliminary data indicate that this FYVE domain can indeed specifically bind to PI(3)P (C. Veigel and F. Zierhut, unpublished observations). So far, nothing is known about the lipid composition of endosomal membranes in trypanosomes, but the presence of a FYVE domain and the enrichment of TbMyo1 at endosomal membranes suggests PI(3)P lipids in the endosomes of trypanosomes too. If so, the expression of the FYVE domain conjugated to GFP might be a useful tool to visualise PI(3)P-enriched membranes in *T. brucei*. It should be stressed that the present study has not rigorously investigated a direct functional role for TbMyo1 in endomembrane function, and any associations that are drawn based on the data here are necessarily speculative. Further experiments using the TbMyo1 RNAi cell line will be needed to explore this point in more detail.

In contrast to the results with the biosynthetic markers, physiological endocytic cargo (BSA, transferrin) visually colocalised with TbMyo1, but the correlations were weak ($\rho$ = 0.17 and 0.20, respectively) (*Figure 3E*). This highlights a potentially important point which is often glossed over in colocalisation analyses – correlation and co-occurrence are different concepts, although both are measures of colocalisation (*Aaron et al., 2018*). Thus, it is possible for two signals to be strongly correlated even if they do not exhibit much spatial overlap/co-occurrence. Conversely, and as here, two signals can exhibit strong overlap/co-occurrence but low correlation. A possible explanation for the low correlation between TbMyo1 and endocytic cargo might be due to topological differences in the signal origin. While TbMyo1 binds to the endosomal membranes from 'outside' (the cytosolic side), the cargo is located in the endosome lumen. Closer examination of some of the data used for these analyses revealed that there are many pixels with a value of 0 for transferrin/BSA and a nonzero value for TbMyo1 and vice-versa. The incidence of zero-versus-nonzero values in the two channels will have lowered the correlation coefficient, and in this sense, the correlation coefficients are giving a hint of what the immuno-electron microscopy later confirmed: that the TbMyo1 and cargo are present in the same locations (i.e. cargo-containing endosomes), but in different proportions (*Figure 4*).

Apart from its endosomal localisation, another intriguing observation stemming from the electron microscopy data was the predominant presence of TbMyo1 signals within the cytoplasm (*Figure 4*). This agrees with the fractionation results (*Figure 1*) and reaffirms that TbMyo1 has a substantial cytosolic portion under steady-state conditions. However, it is important to note that the actin filaments of trypanosomes have not yet been observed by electron microscopy. Recently, advanced techniques like cryo-transmission electron microscopy and cryo-electron tomography have been applied to explore and visualise actin in other cell types (reviewed in *Jung et al., 2020*). It would be intriguing to test these techniques for visualising actin in trypanosomes.

Interestingly, there was also a clear glycosomal localisation of TbMyo1, which was further confirmed by CLEM (*Figure 5C and D*). TbMyo1 was previously found in a glycosomal proteome obtained by

mass spectrometry (see Supplementary Material, Table S2, proteins only detected in BSFs in *Verto-mmen et al., 2008*), providing independent validation of the ultrastructural localisation observed here. Glycosomes are related to peroxisomes and are used to compartmentalise glycolysis in kineto-plastids (*Crowe and Morris, 2021*). While a connection to the cytoskeleton has been described as a requirement for peroxisome function, inheritance, and maintenance in plants, yeast, and mammalian cells (*Neuhaus et al., 2016*), nothing has been reported regarding a connection to the cytoskeleton for glycosomes in trypanosomatids. This link between glycosomes and TbMyo1 was unexpected and merits further investigation.

The results from the CLEM experiments underscored the proximity between TbMyo1 and the lysosome (*Figure 5A and B*). The consistent localisation of TbMyo1 at or close to the endosomal network and lysosomes, coupled with the fact that neither TbMyo1 nor actin accumulated near the flagellar pocket, suggests that the actomyosin system in trypanosomes primarily functions in post-endocytic trafficking events. This encompasses activities related to trafficking within the endosomal system, movement to and from it, or transport to the lysosome. Transport to the lysosome could conceivably be achieved by TbMyo1 interacting with the class II clathrin-coated vesicles that mediate this traffic. Further support for this interpretation comes from the colocalisation studies using antibodies against the endosomal marker proteins TbRab5A, TbRab7, and TbRab11. TbMyo1 exhibited partial overlap with TbRab5A (early endosomes) and TbRab7 (late endosomes), while no overlap was observed with TbRab11 (recycling endosomes). These results support the idea that TbMyo1 is implicated in post-endocytic processes rather than engaging in recycling events.

Actin and TbMyo1 were both previously localised to the posterior region of *T. brucei* using polyclonal antibodies in the 2004 *García-Salcedo et al., 2004* and 2010 *Spitznagel et al., 2010*. studies, respectively, and the latter paper also showed that disruption of actin using either RNAi or LatA resulted in a more diffuse TbMyo1 signal. Until now, however, TbMyo1 and actin had not been imaged simultaneously in *T. brucei*. Here, the actin cytoskeleton was visualised using an inducible anti-actin chromobody, which enabled colabelling with TbMyo1 (*Figure 6*). The ability to observe actin distributions in living trypanosome cells will undoubtedly be invaluable for future studies of its intracellular dynamics, although further optimisation is still required. The overnight inductions used here are undoubtedly excessive, and some cells with abnormal morphology could be observed at low frequency. It will be important in the future to do a time-course experiment to determine the shortest incubation time required for maximal expression of the chromobody. As TbMyo1 is a motor protein, the strong overlap with actin was expected and suggests that actin is highly decorated with TbMyo1. Nevertheless, the number and intensity of signal spots varied from cell to cell. Coupled with the relatively high velocity measured of TbMyo1 on actin, this suggests that the actomyosin system in *T. brucei* is highly dynamic. The obvious next step would be the acquisition of the dynamics of the different endosomal marker proteins and the cytoskeleton in vivo.

One particular topic that would be worth revisiting in light of the data collected here is the role of trypanosome actin in endocytosis. It was surprising that no strong or consistent signal was seen for either TbMyo1 or actin at the flagellar pocket, and it might be possible that neither are essential for endocytosis to occur. Recent work in mammalian cells has suggested that actin is recruited to endo-cytic sites primarily when endocytosis stalls, and a similar mechanism could conceivably be at work here (*Jin et al., 2022*). Regardless, it seems clear that the primary focus and function of trypanosome actin and TbMyo1 is centred on the endosomal system.

After confirming TbMyo1's capability to translocate actin and characterising its localisation and overlap with actin, the final question was whether the actomyosin system played a role in maintaining the intricate and continuous nature of the endosomal network. EP1-GFP-expressing cells have previously served as a reliable tool for endosomal system analysis (*Engstler and Boshart, 2004*; *Link et al., 2023*). In this study, the response of these cells to LatA treatment was analysed to test the actomyosin system's contribution to maintaining the integrity of endosomal membranes. As LatA results in the depolymerisation of the filamentous actin that TbMyo1 binds to, the phenotypic effects observed are due to compromised function of both actin and TbMyo1. Prolonged treatment led to an increasing proportion of cells exhibiting anomalous morphologies, including swelling of the endo-somal membranes, and eventually culminating in a rounded cellular form. The observation of clathrin-coated vesicles at the flagellar pocket as well as the preserved Golgi ultrastructure in the experiment

once again suggest a very precise role for the actomyosin system in *T. brucei*. This role is likely directly involved in maintaining the integrity of the endosomal system in BSF cells.

# Materials and methods

## Key resources table

| Reagent type (species) or resource | Designation | Source or reference | Identifiers | Additional information |
|---|---|---|---|---|
| Genetic reagent (*Lama pacos*) | pAC-TagRFP | ChromoTek (Proteintech) | acr | See 'Materials and methods' for a description of subcloning |
| Antibody | Rabbit anti-TbMyo1(729–1168) polyclonal | This paper | 2411, 2412 | 1:5000-1:10,000 for immunoblotting; 1:500-1:2000 for immunofluorescence |
| Commercial assay or kit | REVERT Total Protein Stain | LI-COR | 926-11011 | Wash and Destaining solutions made in-house |
| Chemical compound, drug | Digitonin, ultra-pure | Merck | CAS 11024-24-1 | See 'Materials and methods' |
| Chemical compound, drug | Latrunculin A | Sigma | CAS 76343-93-6 | See 'Materials and methods' |

## Cell culture and cell lines

*T. brucei* monomorphic BSF parasites (Lister strain 427, antigenic type MITat 1.2, clone 221) were cultured in HMI-9 medium (*Hirumi and Hirumi, 1989*) supplemented with 10% foetal bovine serum (Sigma-Aldrich, USA), 100 U/ml penicillin, and 0.1 mg/ml streptomycin. The single marker (SM) cells (*Wirtz et al., 1999*) were cultivated in the presence of G418 (2.5 µg/ml). For the fractionation experiments, the GFP$^{ESPro}$-221$^{ES}$.121$^{tet}$ cell line was used (*Batram et al., 2014*). This cell line was cultured in the presence of blasticidin (5 µg/ml). Blasticidin was also used for growing the mNG-tagged TbMyo1 and TbMyo21 cells. The TbMyo1 and TbMyo21 RNAi cells were cultivated in the presence of G418 (2.5 µg/ml) and hygromycin (5 µg/ml). The 13–90 cells *Wirtz et al., 1999* used for immunogold experiments were cultivated using hygromycin (5 µg/ml) and G418 (2.5 µg/ml). The transgenic EP1::GFP cells were based on MITat1.2 expressing wild-type cells and cultivated in the presence of hygromycin (5 µg/ml) and G418 (15 µg/ml) (*Engstler and Boshart, 2004*). The cells were maintained at 37°C and 5% $CO_2$ in cell culture flasks with filter lids (Greiner). The cells were split at regular intervals to keep the population density below $1 \times 10^6$ cells/ml. Population density was monitored using a Z2 Coulter Counter (Beckman Coulter).

## Generation of transgenic cell lines

### mNG-tagged cells

The 3xTy1-mNG-3xTy1-tagged TbMyo1 and TbMyo21 cell lines were generated via endogenous replacement using a PCR tagging method in the wild-type strain 427 (*Dean et al., 2015*). For this, the pPOTv6-mNG-blast plasmid was used as the template DNA (*Dean et al., 2015*). This plasmid contains the sequences of the 3xTy1-mNG-3xTy1-tag and a blasticidin resistance gene. The forward primer contained the last 80 bp of the 3' end of the 5'UTR and followed by 20 bp of the sequence just upstream of the blasticidin resistance gene start codon in the SK473 plasmid (from Susanne Kramer, University of Würzburg, based on the pPOTv6-mNG-blast plasmid). The reverse primer contained the first 80 bp of the target gene ORF of the TbMyo1 or TbMyo21 excluding the ATG in reverse complement and the last 18 bp of the 2nd 3xTy1-tag also in reverse complement.

### RNAi cells

These cells were generated in the SM parental cell line (*Wirtz et al., 1999*) using the p2T7_TAblue plasmid (*Alibu et al., 2005*) as a vector. The RNAi target sequences were selected by entering the target gene ORF into the RNAit online tool (*Redmond et al., 2003*). The selected primer pairs flanked the base pairs 286–750 (TbMyo1) or 2137–2603 (TbMyo21) and also encoded homology arms for ligation into p2T7_TAblue. The RNAi target fragments were amplified by PCR from synthesised copies of Tb*MYO1* and Tb*MYO21* in a pUC57 plasmid (Eurogentec). The fragments were ligated into Eam1105I-cut p2T7_TAblue using in vivo assembly (IVA) (*Watson and García-Nafría, 2019*). DNA sequencing was used to confirm correct incorporation of the inserts.

## Actin chromobody cells

For the subcloning of the actin chromobody sequence from the pAC-TagRFP plasmid (ChromoTek, Germany) into the pLew100_v5_Hyg plasmid (*Wirtz et al., 1999*), primers were designed with complementary overhangs for IVA (*Watson and García-Nafría, 2019*). The forward primer overhang sequence was homologous to bases upstream of the HindIII restriction site in the pLew100 plasmid, and the reverse primer overhang was homologous to bases downstream of the BamHI restriction site in the pLew100 plasmid. The second segment of the primers exhibited homology regions with the beginning and end of the actin chromobody sequence, respectively.

## RNAi growth curves

RNAi cells were seeded at $5 \times 10^3$ cells/ml in a volume of 22 ml and divided into two 10 ml aliquots in separate flasks. Tetracycline was added to a final concentration of 1 µg/ml in one flask to induce RNAi and refreshed every 24 hr. The cells were split and reseeded after 48 hr of induction. The population density of the control and induced TbMyo21 RNAi cells was measured every 24 hr over a time course of 96 hr using a Z2 Coulter Counter (Beckman Coulter).

## Differentiation of BSF cells to PCF cells

$2 \times 10^7$ cells were grown to a high density (>$2 \times 10^6$ cell/ml) and harvested by centrifugation (1000 × *g*, 10 min, room temperature [RT]). The supernatant was discarded, and the cell pellet was resuspended in 5 ml DTM (*Vassella and Boshart, 1996*). Here, 1 l DTM medium contains 6.8 g NaCl, 0.4 g KCl, 0.2 g $CaCl_2$, 0.14 g $NaH_2PO_4$, 0.2 g $MgSO_4$, 7.94 g HEPES, 2.2 g $NaHCO_3$, 1 ml phenol red (10 mg/ml), 20 ml MEM amino acids solution 50× (Thermo Fisher, ref no. 11130036), 10 ml MEM non-essential amino acids solution 100× (Thermo Fisher, ref no. 11140035), 1.63 g L-glutamine, 0.293 g L-glutamic acid hydrochloride, 730 µl glycerol, 10 ml MEM vitamin solution 100× (Thermo Fisher, ref no. 11120037), 14 µl *β*-mercaptoethanol, 114 mg Na-pyruvate, 14 mg hypoxanthine, 28.2 mg bathocuproin, 7.5 mg hemin, 640 mg proline, and 182 mg cysteine. The medium was prepared using $ddH_2O$ and adjusted to pH 7.5. Afterwards, 150 ml heat-inactivated (1 hr, 56°C) foetal calf serum (Sigma-Aldrich) was added, and the medium was sterilised by filtration. After the addition of *cis*-aconitate to a final concentration of 6 mM, the cells were transferred into cell culture flasks and incubated overnight at 27°C. Most cells had PCF morphology the next morning and could be cultivated in SDM79 media (*Brun, 1979*) from that point onwards.

## Expression and purification of full-length TbMyo1

Full-length TbMyo1 was cloned into the pFastBac1 expression vector, flanked by a N-terminal 6xHIS-tag for affinity purification and a C-terminal AviTag to allow biotinylation of the protein. The myosin heavy chain was co-expressed in the presence of human calmodulin using a baculoviral/Sf21 system (*Batters et al., 2014*). All remaining steps were performed at 4°C. The pellet of 500 ml Sf21 cells was resuspended in lysis buffer (20 mM HEPES pH 7.4, 300 mM NaCl, 2 mM $MgCl_2$, 1 mM DTT, 2 mM ATP, 2 µM calmodulin, and EDTA-free protease inhibitor tablets [Roche]) and lysed by sonication (sonicator probe Bandelin HD 2070, 5 min total, at 50% maximal power). The cell extract was centrifuged at 20,000 rpm in a JA 25.50 (Beckman) rotor for 30 min. The filtered lysate was then incubated for 30 min with 1 ml of equilibrated HIS-Select HF Ni-affinity gel (Sigma) and loaded onto a gravity flow column. The resin was washed five times with 3 ml of wash buffer (20 mM HEPES, 150 mM NaCl, 1 mM $MgCl_2$, 1 mM DTT, pH 7.4). Subsequently, myosin was eluted with 150 mM imidazole in wash buffer. All elution fractions were pooled, concentrated in a spin concentrator (50k MWOC Amicon), and biotinylated using the BirA enzyme (*Gautier and Hinner, 2015*). As a final purification step, the biotinylated myosin sample was loaded onto a Superdex 200 10/300 column to perform size-exclusion chromatography. Peak fractions were analysed using SDS-PAGE, pure fractions were pooled and flash-frozen in liquid nitrogen. The protein was stored at –80°C.

## Anti-TbMyo1 antibodies

Anti-TbMyo1 antibodies were generated by immunisation of two rabbits with purified recombinant TbMyo1 (amino acids 729–1168) (Eurogentec). For recombinant protein expression, the TbMyo1 tail was cloned into a pET-28a(+) vector including an N-terminal 6xHIS-tag. The protein was expressed in BL21 (DE3) *Escherichia coli* cells for 4 hr at 27°C. Expression was induced with 1 mM of IPTG.

Cells were harvested by centrifugation and dissolved in 30 ml of lysis buffer (50 mM Tris–HCl pH 7.5, 500 mM NaCl, 40 mM imidazole, 1 mM DTT) and lysed by sonication (sonicator probe Bandeline HD 2070) for 5 min at 50% maximal power. The lysate was centrifuged at 20,000 rpm for 20 min at 4°C in a JA 25.50 rotor (Beckman) and the supernatant was filtered through a 0.45 µm syringe filter. Subsequently, the supernatant was loaded onto an equilibrated 5 ml HisTrapFF column (GE Healthcare) using an ÄKTA pure. After washing the column with lysis buffer, proteins were eluted using 200 mM imidazole in lysis buffer. Pure samples were pooled and concentrated using a spin concentrator (10K MWOC Amicon). As a final step, size-exclusion chromatography was conducted. The protein sample was applied to a Superdex 200 10/300 GL column which was equilibrated in PBS, pure elution fractions were pooled, flash frozen in liquid nitrogen, and stored at –80°C. After validation of specificity, the antisera were mixed in a 2:1 ratio and used as a cocktail in further experiments.

## Previously published and commercial antibodies

The following primary antibodies have been described previously: mouse anti-Ty1 ('BB2') (*Bastin et al., 1996*), mouse anti-PFR1,2 ('L13D6') (*Kohl et al., 1999*), rabbit anti-GFP (*Pelletier et al., 2002*), rabbit anti-TbBiP (*Bangs et al., 1993*), mouse anti-p67 (*Alexander et al., 2002*), rabbit anti-GRASP (*He et al., 2004*), rat anti-TbRab5A, guinea pig anti-TbRab7, and guinea pig anti-TbRab11 (*Link et al., 2023*).

The following antibodies came from commercial sources: goat anti-rabbit (IRDye800CW, LI-COR), goat anti-mouse (IRDye680LT, LI-COR), goat anti-rabbit and anti-mouse antibodies conjugated to AlexaFluor dyes (Molecular Probes).

## Biochemical fractionation

The cell concentration of the GFP$^{ESPro}$-221$^{ES}$.121$^{tet}$ and SM cell lines was measured using a Z2 Coulter Counter (Beckman Coulter). The two cell lines were used in a 2:1 ratio to dilute the GFP signal and transferred to 50 ml Falcon tubes and pelleted by centrifugation (1000 × $g$, 10 min, RT). The supernatant was removed, and each cell pellet was resuspended in 500 µl vPBS and pooled into one single microfuge tube. The washed cells were pelleted again by centrifugation (1000 × $g$, 2 min, RT). A second wash step with 1 ml vPBS was then carried out. After the second wash, the cell pellet was resuspended in 400 µl of 40 µg/ml ultra-pure digitonin (Merck, CAS 11024-24-1) in vPBS and incubated (25 min, RT). A 5% (20 µl) input (I) sample was then taken. The mixture was then separated by centrifugation (750 × $g$, 5 min, RT) and 320 µl of the cytosolic fraction (SN1) was transferred to a fresh microfuge tube where a 5% sample was taken. The cell pellet was then resuspended with 1 ml vPBS, and the extracted cells were again pelleted by centrifugation (750 × $g$, 5 min, RT). Afterwards, the extracted cells were resuspended in 400 µl lysis buffer (1% IGEPAL, 0.1 M PIPES-NaOH pH 6.9, 2 mM EGTA, 1 mM MgCl$_2$, 0.1 mM EDTA, EDTA-free protease inhibitor) and incubated 15 min at RT with 500 rpm agitation. Afterwards, a 5% sample (P1) was taken, and the lysed cells were pelleted by centrifugation (3400 × $g$, 2 min, RT). Next, 320 µl of the supernatant (SN2) was transferred to a fresh microfuge tube where a 5% sample was taken. The cytoskeletal pellet was resuspended in 1 ml vPBS containing EDTA-free protease inhibitors (Roche) and centrifuged again (3400 × $g$, 2 min, RT). The pellet (P2) was resuspended in 400 µl extraction buffer and a 5% sample was taken. SDS loading buffer was added to the 5% samples (I, SN1, P1, SN2, P2) to a final volume of 40 µl. The samples were boiled at 100°C for 10 min and stored at – 20°C prior to use.

## Immunoblotting

For the preparation of whole-cell lysates (WCL), cells were harvested by centrifugation (1000 × $g$, 10 min) and the cell pellet was resuspended in 1 ml vPBS containing EDTA-free protease inhibitor (Roche). The cells were pelleted again by centrifugation (750 × $g$, 3 min, RT) and the supernatant was removed. SDS loading buffer was then added to a final concentration of 2 × 10$^5$ cells/µl. The samples were further denatured by boiling (100°C, 10 min) and stored at –20°C until use. SDS-PAGE was carried out using a Mini-PROTEAN Tetra cell (Bio-Rad) and transfer to nitrocellulose membranes was done using a mini-Trans blot cell (Bio-Rad), both according to the manufacturer's instructions. Protein transfer and equal loading were confirmed using a REVERT total protein stain (LI-COR) according to the manufacturer's instructions. Membranes were blocked using blocking buffer (PBS, 0.3% [v/v] Tween 20, 10% [w/v] milk) (30 min, RT, rocker). The membranes were then incubated in primary antibodies

diluted in blocking buffer (1 hr, RT, roller). After three washes in immunoblot buffer (PBS, 0.3% Tween 20), the membranes were incubated with IRDye-conjugated secondary antibodies diluted in immunoblot buffer (1 hr, RT, roller). After another three washes in immunoblot buffer, the membranes were visualised using an Odyssey CLx (LI-COR).

## Actin filament gliding assay

Motility assay procedures were adapted from those described by *Kron et al., 1991*. Clean glass coverslips 24 × 50 mm (VWR) were functionalised with a mixture of mPEG-silane MW 2000 and biotin-PEG-silane MW 3400 (Laysan Bio). In brief, both were dissolved in 80% ethanol pH 2.0 and mixed in a 20:1 ratio. 100 µl of the PEG-biotin solution was spread on each coverslip and incubated for 30 min at 70°C. Afterwards, the coverslips were washed with ethanol, dried under an air stream, and flow chambers were assembled. C-terminal biotinylated myosin was specifically bound via streptavidin to the functionalised glass surface. Subsequently, the surface of the experimental chamber was blocked with 0.5 mg/ml BSA in assay buffer (25 mM imidazole, 25 mM KCl, 4 mM $MgCl_2$, 1 mM EGTA, pH 7.4) for 3 min. After washing the chamber with assay buffer, Alexa488-phalloidin-labelled rabbit skeletal actin filaments were introduced into the flow cell. The assay was started by adding 2 mM ATP in the assay buffer containing a scavenger system (10 mM DTT, 0.01 mg/ml catalase, 0.05 mg/ml glucose oxidase, 1.5 mg/ml glucose) to prevent photobleaching of the fluorescent actin filaments. Actin filaments were recorded by fluorescence imaging using a 488 nm laser (40 mW) and images were taken every 1 s with an EMCCD (Andor iXon3) camera using a Nikon Ti-Eclipse microscope in TIRF mode. Moving filaments were tracked and analysed using the ImageJ plugin MTrackJ and Origin.

## Preparation of samples for immunofluorescence microscopy

Coverslips were washed in 70% ethanol and then coated with 0.01% poly-L-lysine in a 24-well plate (>20 min, RT) and left to dry. $2 \times 10^6$ cells were taken per coverslip and transferred to 15 ml Falcon tubes. The cells were pelleted by centrifugation (1000 × $g$, 1 min per ml of liquid, RT) in a swing-bucket centrifuge. The supernatant was removed, and the cell pellet was gently resuspended in 1 ml ice-cold vPBS containing EDTA-free protease inhibitor (Roche). The cells were again pelleted by centrifugation (1000 × $g$, 2 min, RT) and subsequently resuspended in 0.5 ml ice-cold vPBS containing EDTA-free protease inhibitor (Roche). 0.5 ml ice-cold 8% (v/v) paraformaldehyde solution in vPBS containing EDTA-free protease inhibitor (Roche) or ice-cold 8% (v/v) paraformaldehyde solution/0.2% (v/v) glutaraldehyde in vPBS containing EDTA-free protease inhibitor (Roche) was added to the resuspended cells. Glutaraldehyde was omitted if a subsequent antibody labelling was performed. The suspension was mixed 1–2 times by inversion and was placed on ice for 10 min. The cells were then moved to RT and were incubated for a further 30 min. Afterwards, the fixed cells were centrifuged at 1000 × $g$ for 2 min at RT. The supernatant was removed, and the fixed cells were then resuspended in 1 ml RT vPBS containing EDTA-free protease inhibitor (Roche). The cells were attached to the coverslips by centrifugation (1000 × $g$, 1 min, RT); attachment was confirmed visually. If the cells were fixed with 4% (v/v) paraformaldehyde solution and 0.1% (v/v) glutaraldehyde in vPBS containing EDTA-free protease inhibitor (Roche), the coverslips were directly mounted on glass slides after cell attachment. Cells which were fixed with 4% paraformaldehyde solution in vPBS containing EDTA-free protease inhibitor (Roche) were first incubated with 0.25% (v/v) TritonX-100 in PBS for 5 min at RT to permeabilise the cells. The coverslips were blocked in 1 ml 3% (w/v) BSA in PBS (30 min, RT), and sequentially incubated with clarified primary and secondary antibodies diluted in PBS (1 h, RT, humidified chamber for each) with three PBS washing steps (3 × 5 min, RT, rocker) after each incubation. After the final wash, glass slides were cleaned with 70% ethanol and a spot of DAPI-Fluoromount (Southern Biotech) was placed on the surface. The coverslips were rinsed in $ddH_2O$, carefully dried, and then mounted. For preparation of samples for SIM imaging, different coverslips (high precision, No. 1.5 H) were used. Cells were harvested as above, washed, and fixed in 4% paraformaldehyde solution (10 min, ice then 30 min, RT). The fixed cells were washed, attached to coverslips, permeabilised (0.25% Triton X-100 in PBS; 5 min, RT), and then labelled and mounted as described above.

## Expansion microscopy

Cells were initially prepared in the same way as for immunofluorescence microscopy. After the incubation with secondary antibodies and washing, 1 ml 25 mM MA-NHS (bifunctional cross-linker, prepared

immediately prior to use) was added to the coverslips (*Truckenbrodt et al., 2019*). The coverslips were then incubated on a rocker (30 min, RT). MA-NHS was removed, and the coverslips were washed with 1 ml PBS. To allow cells to equilibrate in the monomer solution (8.625% [w/v] sodium acrylate, 20% [w/v] acrylamide, 0.075% [w/v] bisacrylamide, 2 M NaCl in PBS), 1 ml was added and incubated for 5 min at RT. 100 µl of monomer solution, supplemented with 2 µl 10% (v/v) APS and 2 µl 10% (v/v) TEMED, was briefly vortexed and a 70 µl drop was placed on a piece of parafilm. Coverslips were transferred with cells facing down onto the drop. The polymerisation was allowed to proceed for 20 min at RT in the dark. The resulting gel was peeled off the parafilm and transferred into a 12-well plate, with the coverslip at the bottom. For digestion, 1 ml digestion buffer (0.8 M guanidine HCl, 0.5% [v/v] Triton X-100, 1× TAE buffer) was mixed with 10 µl proteinase K (6 U/ml) and added to gels. Digestion was carried out at 37°C overnight in the dark. Gels were transferred into 10 cm Petri dishes. For expansion, ddH$_2$O (~50 ml) was added until the gels were completely covered. The gels were incubated in the dark at RT for 3 hr. The water was changed every hour. To stain nucleic acids, gels were placed in water containing DAPI (Thermo Fisher Scientific, 5 mg/ml) at a 1:1000 dilution and incubated for 30 min in the dark. Water was removed as much as possible, and the diameters of the gels were measured with a ruler to estimate expansion. Gels were cut into rectangular pieces using a razor blade (~1 cm$^2$) and transferred into an imaging chamber. A 35 mm glass-bottom dish was used as an imaging chamber (Glass Bottom Dish, Thermo Fisher Scientific). The dish was first cleaned with 100% EtOH. The bottom was covered with 1 ml 0.1% (v/v) poly-L-lysine and incubated for 1 hr. Chambers were allowed to air-dry after the lysine was removed.

## Fluorescence microscopy

Images were acquired using a DMI6000B widefield microscope (Leica Microsystems, Germany) with a HCX PL APO CS objective (×100, NA = 1.4, Leica Microsystems) and Type F Immersion Oil (refractive index = 1.518, Leica Microsystems). The microscope was controlled using LAS-X software (Leica). Samples were illuminated with an EL6000 light source (Leica) containing a mercury short-arc reflector lamp (HXP-R120W/45C VIS, OSRAM, Germany). Excitation light was selected by using Y3 (545/25 nm), GFP (470/40 nm), and A4 (360/40 nm) bandpass filter cubes (Leica Microsystems). The power density, measured at the objective focal plane with a thermal sensor head (S175C, Thorlabs), was respectively 0.749 ± 0.086, 0.557± 0 .069, and 0.278 ± 0.076 W/cm$^2$ for the three filters. Emitted light was collected at ranges of 605/70 (Y3), 525/50 nm (GFP), and 470/40 nm (DAPI). The individual exposure times and camera gains were adjusted according to the different samples. The mNG-tagged samples required extremely long exposure times (2 s per z-section). RNAi samples (induced and control) were imaged using identical settings. Differential interference contrast (DIC) was used to visualise cell morphology. 3D recordings of each field of view were obtained using 40 Z-slices (step size = 0.21 µm). Fields of view were selected in the DIC channel in order to blind the user to the fluorescence signal and subjectively select for cells with optimum morphology. Images were captured using a DFC365 FX monochrome CCD camera (Leica, 6.45 µm pixel size). SIM images were acquired using an Elyra S.1 SIM microscope (×63 oil objective, NA = 1.4, Zeiss, Germany) controlled by ZEN software (Zeiss). The microscope was equipped with a sCMOS camera PCO Edge 5.5 (Excelitas PCO GmbH, Germany).

## Cargo uptake assays
### BSA uptake

Cells (2 × 10$^6$ cells per coverslip for widefield microscopy, 1 × 10$^8$ for electron microscopy) were harvested by centrifugation (1000 × *g*, 10 min, 4°C) and washed in 1 ml ice-cold Voorheis' modified PBS (vPBS; PBS, 46 mM sucrose, 10 mM glucose). The washed cells were pelleted by centrifugation (750 × *g*, 2 min, 4°C), and resuspended in 100 µl ice-cold vPBS. The cells were then incubated at low temperature (10 min, ice) to block endocytosis. During this incubation, the labelled BSA (AlexaFluor555 conjugate, Molecular Probes; Alexa555-BSA-Au, UMC Utrecht) was clarified by centrifugation (750 × *g*, 1 min, RT) and kept on ice in the dark. While for widefield microscopy experiments Alexa555-BSA was added to a final concentration of 600 µg/ml, for electron microscopy experiments Alexa555-BSA-5nm-gold conjugate was added to a final OD = 5. Afterwards, the samples were incubated at low temperature (15 min, on ice, in the dark) to allow the BSA to enter the flagellar pocket. The following temperature shift and additional incubation allowed the endocytosis of BSA (15 min or 30 min, 37°C, in the dark). To quench the reaction, 1 ml ice-cold vPBS was added. The cells were

pelleted by centrifugation (750 × $g$, 2 min, 4°C) and resuspended in 50 μl vPBS containing EDTA-free protease inhibitor (Roche). The cells were fixed by addition of 0.5 ml ice-cold fixation solution (4% [v/v] paraformaldehyde solution and 0.1% [v/v] glutaraldehyde in vPBS for widefield microscopy, 2% [v/v] paraformaldehyde solution and 0.2% [v/v] glutaraldehyde for electron microscopy) (20 min, on ice, in the dark), followed by a 30 min (widefield microscopy) or 2 hr (electron microscopy) incubation at RT (in the dark). Afterwards, the cells were pelleted by centrifugation (750 × $g$, 2 min, RT) and washed in 4 ml vPBS containing EDTA-free protease inhibitor (Roche). The washed cells were again pelleted by centrifugation (750 × $g$, 4 min, RT), resuspended in 1 ml vPBS containing EDTA-free protease inhibitor (Roche), and – for widefield microscopy – attached to the poly-L-lysine-coated coverslips by centrifugation (750 × $g$, 1 min, RT). The coverslips were then mounted on glass slides using DAPI-Fluoromount (Southern Biotech) and imaged immediately. For electron microscopy, the cells were further treated as explained in the sample embedding and immuno-electron microscopy section.

### Transferrin uptake

In general, most preparation steps were performed in the same way as for BSA uptake (see above). After the first washing step, however, the cells were resuspended in 1 ml RT vPBS and incubated (15 min, 37°C) to allow the internalisation of surface-bound transferrin from the cell culture media. Afterwards, the cells were pelleted by centrifugation (1000 × $g$, 2 min, 4°C) and resuspended in 50 μl ice-cold vPBS, then incubated at low temperature (10 min, on ice) to block endocytosis. After this incubation time, 0.5 μl of 5 mg/ml transferrin (Tetramethylrhodamine conjugate, ThermoFisher Scientific) was added, mixed by flicking, and incubated (15 min, on ice, in the dark) to allow the transferrin to enter the flagellar pocket. The subsequent temperature shift (30 min, 37°C, in the dark), fixation, washing step, attachment to the coverslips, and mounting onto the glass slides were performed as described in the BSA uptake method (see above). The samples were imaged immediately.

## Sample embedding for electron microscopy

For Tokuyasu immuno-EM, sample preparation and sectioning were performed as previously described (*Link et al., 2023*). In brief, 1 × 10^8 cells were harvested by centrifugation (1500 × $g$, 10 min, 37°C) and washed with 1 ml trypanosome dilution buffer (TDB; 5 mM KCl, 80 mM NaCl, 1 mM MgSO$_4$, 20 mM Na$_2$HPO$_4$, 2 mM NaH$_2$PO$_4$, 20 mM glucose). Cells were fixed by adding 4% (v/v) formaldehyde and 0.4% (v/v) glutaraldehyde in 0.2 M PHEM buffer (120 mM PIPES, 50 mM HEPES, 4 mM MgCl$_2$, 20 mM EGTA, adjusted to pH 6.9 with 1 M NaOH) 1:1 to TDB. After 5 min, the fixatives were replaced with 2% (v/v) formaldehyde and 0.2% (v/v) glutaraldehyde in 0.1 M PHEM for 2 hr at RT. Fixatives were removed by washing and quenched with PBS + 0.15% (w/v) glycine. Cells were pelleted and resuspended in 200 μl 12% gelatine (Rousselot 250 LP30) in 0.1 M PHEM at 37°C and pelleted again. The pellets were frozen on ice, cut into smaller blocks, and infused with 2.3 M sucrose overnight at 4°C. The infused blocks were mounted on metal pins and stored in liquid nitrogen. The gelatine-embedded cells were cryosectioned to 55 nm or 100 nm thick sections at –120°C or –100°C, respectively, using a DiATOME diamond knife in a Leica ultracut cryomicrotome (UC7). Sections were picked up and deposited on formvar- and carbon-coated grids using 2.3 M sucrose and 1.8% (v/v) methylcellulose (MC, Sigma M6385 25 centipoises) mixed 1:1.

For high-pressure freezing (HPF), cells were grown to a density of 1 × 10^6 cells/ml in 50 ml, incubated with LatA for 30–60 min and harvested by centrifugation (1000 × $g$, 10 min, RT). The supernatant was removed to 4 ml, and 4 ml foetal bovine serum was added. The cells were pelleted again (1000 × $g$, 10 min, RT) and the supernatant was removed to 200 μl. The cells were resuspended in the supernatant, and the suspension was then transferred to a PCR tube and the cells pelleted by centrifugation (1600 × $g$, 10 s, RT). The cells were then transferred into a carrier with a closed lid to avoid air inclusions. HPF was started immediately using a Leica EM HPM100. After HPF, the samples were transferred to an advanced freeze substitution machine (Leica EM AFS2). Low-temperature embedding and polymerisation of Epon raisin [DDSA, MNA, Epon812, 2,4,6 Tris(dimethylaminomethyl(phenol))] were then carried out. Ultra-thin cuts (60 nm) were obtained with an ultramicrotome (Leica EM UC7/FC7) and were placed on slotted grids. For contrasting, the slices were incubated in 2% uranyl acetate (UA) for 10 min. Afterwards, the grids were washed 3× in ddH$_2$O (boiled to remove CO$_2$) and incubated for 5 min on 50% Reynold's lead citrate in a Petri dish with NaOH tablets. The grids were again washed 2× in ddH$_2$O, dried and stored at RT until imaging.

## Immuno-electron microscopy

Grids with sections facing down were incubated in a 24-well plate with 2 ml PBS at 37°C for 30–60 min to remove gelatine, 2.3 M sucrose, and 1.8% (v/v) MC mixture. Afterwards, the grids were washed on ~100 µl drops of PBS + 0.15% (w/v) glycine and PBS on parafilm at RT (both 3 × 5 min). 100 µl drops represent the standard volume for all non-antibody related steps. Next, blocking was performed in PBS + 0.1% (v/v) acetylated BSA (BSA-c) + 0.5% (w/v) cold fish-skin gelatine (FSG) for 30 min. The following primary antibody incubation was done for 1–2 hr in PBS + 0.1% BSA-c + 0.5% FSG. Afterwards, grids were washed 5 × 5 min with PBS + 0.1% BSA-c + 0.5% FSG and incubated with corresponding gold or fluorophore-coupled secondary antibodies for 1–2 hr. Further washing steps with PBS + 0.1% BSA-c + 0.5% FSG (3 × 5 min) and PBS (3 × 5 min) followed before all remaining antibodies were fixed with 1.25% glutaraldehyde for 5 min at RT. After three washing steps with ddH$_2$O to remove phosphate from the grids, the grids were differently treated if an immunogold or a CLEM assay was performed. For CLEM assays, the grids were mounted with ProLong Diamond Antifade with DAPI (Life Technologies) between a coverslip and glass slide and imaged using a DMI6000B widefield microscope as described above. Afterwards, the grids were unmounted and washed five times with ddH$_2$O. Then (for immunogold and CLEM assay), the sample contrast was increased by 5 min incubation on 2% (v/v) UA in water (pH 7) and 7 min incubation on 1.8% (v/v) methyl cellulose (MC)/0.3% (v/v) UA in water (pH 4) (both on ice). Grids were looped out with a metal ring and excess MC/UA was removed with filter paper. Grids were dried for at least 30 min at RT in the metal ring. Grids treated in this way can be stored indefinitely and imaged on an electron microscope when desired.

## Electron microscope

A JEOL JEM-1400 Flash scanning transmission electron microscope (STEM) was operated at 120 kV with camera system: Matataki Flash 2k × 2k.

## Flow cytometry

Tetracycline-induced and -uninduced cells (1 × 10$^6$ per sample) were harvested (1500 × $g$, 3 min, RT), resuspended in 1 ml filtered vPBS, and stored on ice until usage. The fluorescence intensity measurements were carried out with FACSCalibur (BD Bioscience). For the measurement of the fluorescence intensity of the samples, the fluorescence channel FL2-H (filter range 564–601 nm) was selected to detect the fluorescence intensity of anti-actin TagRFP signal (detection wavelength 588 nm). The data was visualised using the flow cytometry software BD CellQuest Pro.

## Latrunculin A treatment

Uninduced and induced samples were treated with either 0.2 µM or 2 µM LatA in DMSO (Sigma, CAS 76343-93-6) up to 4 hr at 37°C. After the Latrunculin treatment, untreated and treated cells were fixed using 4% (v/v) paraformaldehyde solution and 0.1% (v/v) glutaraldehyde in vPBS, attached to coverslips, and mounted onto the glass slides as described in the preparation for immunofluorescence microscopy section. The samples were imaged immediately.

## Data analysis

### Image processing

The microscopy images were processed using FiJi (*Schindelin et al., 2012*) and a macro was used to generate maximum-intensity projections in a semi-automated workflow (Tim Krüger, University of Würzburg).

### Quantification analysis

The immunoblotting results were qualitatively analysed using Image Studio Lite (LI-COR Biosciences). Background subtraction and signal measurement were carried out using Empiria Studio (LI-COR Biosciences). Normalisation against the Total Protein Stain and quantification were completed using Microsoft Excel. Dot plots were created using the web app PlotsOfData (*Postma and Goedhart, 2019*).

### Correlation analysis

The correlation analysis was done using FiJi (*Schindelin et al., 2012*) and a macro for preprocessing the raw data. For this, single cells were selected by using the raw data. Only 1K1N cells were chosen

by using the DIC and the blue channel which shows the nucleus and the kinetoplast. The cells were cropped by a square of 130 × 130 pixels and saved as .tif files. Afterwards, a standardised background noise reduction of the red and green channel using the BioVoxxel 'Convoluted Background Subtraction' plugin was performed. Macros used for opening Bio-Format files were modified from Dr. Jan Brocher/BioVoxxel. Firstly, sum slices projections were prepared and converted to 8-bit images. For the subtraction, the median convolution filter with specific radii was chosen, the radius was selected so that only the signal with the highest intensity was visible. The two channels (green and red) were saved in separate folders. A 2D colocalisation macro was carried out on these images. This macro was adapted from the one designed by *Zhang and Cordelières, 2016*. Based on the cytofluorograms generated by this macro, the correlation method was chosen as the Spearman's rank. The Spearman's rank correlation coefficient of each image and marker was copied into an Excel spreadsheet. A dot plot of the results was generated using GraphPad Prism.

## CLEM analysis

The ec-CLEM plug-in in Icy was used for the initial correlation (*Paul-Gilloteaux et al., 2017*). Adobe Photoshop was used to generate the final overlay of the images.

## Materials availability statement

The materials used in this study are at the Department of Cell and Developmental Biology, University of Würzburg. The authors commit to making them available upon request.

## Acknowledgements

Thanks go to the Cell Microscopy Core of UMC Utrecht for their workshop 'Cryosectioning and immuno-electron microscopy' and training Fabian Link in the Tokuyasu technique. Katy Schmidt (University of Vienna) provided feedback regarding electron microscopy techniques. Elisabeth Meyer-Natus, Daniela Bunsen, Claudia Gehring-Höhn, and Christian Stigloher provided help with the electron microscopy benchwork and assistance with the electron microscope. Nicolas Hagedorn assisted with the flow cytometry experiments. Chris de Graffenried (Brown University) provided protocols and advice on expansion microscopy. Derek Nolan (Trinity College, Dublin) very generously shared unpublished data on TbMyo21 and provided feedback and advice. Thanks also go to Tom Beneke, Nicola Jones, and Susanne Kramer for proofreading and providing feedback on the manuscript. Non-commercial antibodies were obtained from the following sources: Cynthia He (National University of Singapore) provided the anti-Ty1 (BB2) antibodies; Keith Gull (University of Oxford) provided the anti-PFR1,2 (L13D6) antibodies, and James Bangs (University of Buffalo) provided the anti-p67 and anti-BiP antibodies. AB was supported by the Brazilian agency CAPES (program: CAPES/DAAD-Call No. 22/2018; process 88881.199683/2018–01). ME was supported by DFG grants EN305, SPP1726 (Microswimmers—From Single Particle Motion to Collective Behaviour), GIF grant I-473-416.13/2018 (Effect of extracellular *Trypanosoma brucei* vesicles on collective and social parasite motility and development in the tsetse fly) and GRK2157 (3D Tissue Models to Study Microbial Infections by Obligate Human Pathogens), the EU ITN Physics of Motility, and the BMBF NUM Organo-Strat. ME is a member of the Wilhelm Conrad Röntgen Center for Complex Material Systems (RCCM). CV was supported by DFG grants SFB 863 (B6), SFB 1032 (B1), and Friedrich Baur Stiftung. The transmission electron microscope was funded by the Deutsche Forschungsgemeinschaft (DFG, German Research Foundation)—426173797 (INST 93/1003-1 FUGG). This work was funded by remaining money from DFG grant MO 3188/2-1 to BM.

## Additional information

### Funding

| Funder | Grant reference number | Author |
|---|---|---|
| CAPES/DAAD | Call No. 22/2018 process 88881.199683/2018-01 | Alyssa Borges |

| Funder | Grant reference number | Author |
|---|---|---|
| Deutsche Forschungsgemeinschaft | EN305 | Markus Engstler |
| BMBF | BMBF NUM Organo-Strat | Markus Engstler |
| Deutsche Forschungsgemeinschaft | SFB 863 (B6) | Claudia Veigel |
| Deutsche Forschungsgemeinschaft | SFB 1032 (B1) | Claudia Veigel |
| Friedrich Baur Stiftung | | Claudia Veigel |
| Deutsche Forschungsgemeinschaft | MO 3188/2-1 | Brooke Morriswood |
| Deutsche Forschungsgemeinschaft | SPP1726 | Markus Engstler |
| Deutsche Forschungsgemeinschaft | GIF grant I-473-416.13/2018 | Markus Engstler |
| Deutsche Forschungsgemeinschaft | GRK2157 | Markus Engstler |
| Deutsche Forschungsgemeinschaft | ITN Physics of Motility | Markus Engstler |

The funders had no role in study design, data collection and interpretation, or the decision to submit the work for publication.

## Author contributions

Fabian Link, Conceptualization, Data curation, Formal analysis, Investigation, Visualization, Methodology, Writing – original draft, Writing – review and editing; Sisco Jung, Xenia Malzer, Felix Zierhut, Antonia Konle, Christopher Batters, Johannes Kullmann, Formal analysis, Investigation, Methodology; Alyssa Borges, Software, Formal analysis, Investigation, Methodology, Writing – original draft, Writing – review and editing; Monika Weiland, Mara Poellmann, An Binh Nguyen, Investigation, Methodology; Claudia Veigel, Brooke Morriswood, Conceptualization, Resources, Data curation, Formal analysis, Supervision, Funding acquisition, Validation, Investigation, Visualization, Methodology, Writing – original draft, Project administration, Writing – review and editing; Markus Engstler, Conceptualization, Resources, Data curation, Formal analysis, Supervision, Funding acquisition, Methodology, Writing – original draft, Project administration, Writing – review and editing

## Author ORCIDs

Fabian Link ⓘ https://orcid.org/0000-0002-9828-2012
Christopher Batters ⓘ https://orcid.org/0009-0007-4555-7901
Markus Engstler ⓘ https://orcid.org/0000-0003-1436-5759
Brooke Morriswood ⓘ https://orcid.org/0000-0001-7031-3801

Reviewer #1 (Public Review): https://doi.org/10.7554/eLife.96953.3.sa1
Reviewer #2 (Public Review): https://doi.org/10.7554/eLife.96953.3.sa2
Reviewer #3 (Public Review): https://doi.org/10.7554/eLife.96953.3.sa3
Author response https://doi.org/10.7554/eLife.96953.3.sa4

# Additional files

## Supplementary files
• MDAR checklist

## Data availability

The source data files contain the raw and annotated gel and immunoblot data and numerical data files used for graphs. Newly generated code used in this study can be obtained on https://github.com/alyssaborges/PhD-Thesis_2023 (copy archived at *Borges, 2023*). Due to its size (~300 GB)

and complexity, the raw imaging data is currently not available on a public repository. Interested researchers can request the raw imaging data by contacting the corresponding author; no project proposal or application is required, all reasonable requests will be approved, and there are no restrictions on data access.

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
