## [Editor Report · eLife assessment]

This **important** study builds on a previous publication, demonstrating that *T. brucei* has a continuous endomembrane system, which probably facilitates high rates of endocytosis. Using a range of cutting-edge approaches, the authors present **compelling** evidence that an actomyosin system, with the myosin TbMyo1 as an active molecular motor, is localized close to and can associate with the endosomal system in the bloodstream form of *T. brucei*. It shows **convincingly** that both actin and Myo I play a role in the organization and integrity of the endosomal system: both RNAi-mediated depletion of Myo1, and treatment of the cells with latrunculin A resulted in endomembrane disruption. This work should be of interest to cell biologists and microbiologists working on the cytoskeleton, and unicellular eukaryotes.

---

## [Referee Report · Reviewer #1 (Public Review)]

Using a combination of cutting-edge high-resolution technologies (expansion microscopy, SIM, and CLEM) and biochemical approaches (in vitro translocation of actin filaments, cargo uptake assays, and drug treatment), the authors revisit and update previous results about TbMyo1 and TbACT in the bloodstream form (BSF) of *Trypanosoma brucei*. They show that a great part of the myosin motor is cytoplasmic but the fraction associated with organelles is in proximity to the endosomal system and in glycosomes. In addition, they show that TbMyo1 can move actin filaments in vitro and visualize for the first time this actomyosin system using specific antibodies, a "classical" antibody for TbMyo1, and a chromobody for actin. Finally, using latrunculin A, which sequesters G-actin and prevents F-actin assembly, the authors show the delocalization and eventually the loss of the filamentous actin signal and the concomitant loss of the endosomal system integrity.

Overall this well-conducted and convincing study paves the way toward the elucidation of the role of an actomyosin system in the maintenance of the endosomal network in *T. brucei*.

Strengths:

The work is of high quality and uses advanced technologies to determine the involvement TbMyo1 and actin in the integrity of the endosomal system. The conclusions are not over-interpreted and are supported by the experimental results and their quantification.

Weaknesses:

Although disruption of the actomyosin system using either the actin-depolymerizing drug latrunculin A or the TbMyo1-RNAi cell line established an effect on the endosomal system integrity, it remains to understand how this occurs mechanistically and what are the intracellular components involved.

---

## [Referee Report · Reviewer #2 (Public Review)]

The study by Link et al. advances our understanding of the actomyosin system in *T. brucei*, focusing on the role of TbMyo1, a class I myosin, within the parasite's endosomal system. Using a combination of biochemical fractionation, in vitro motility assays, and advanced imaging techniques such as correlative light and electron microscopy (CLEM), this paper demonstrates that TbMyo1 is dynamically distributed across early and late endosomes, the cytosol, is associated with the cytoskeleton, and a fraction has an unexpected association with glycosomes. Notably, the study shows that TbMyo1 can translocate actin filaments at velocities suggesting an active role in intracellular trafficking, potentially higher than those observed for similar myosins in other cell types. This work not only elucidates the spatial dynamics of TbMyo1 within *T. brucei* but also suggests its broader involvement in maintaining the complex architecture of the endosomal network, underscoring the critical role of the actomyosin system in a parasite that relies on high rates of endocytosis for immune evasion.

A key strength of the study is its exceptional rigor and successful integration of a wide array of sophisticated techniques, such as in vitro motility assays, and advanced imaging methods, e.g. CLEM. This combination of approaches underscores the study's comprehensive approach to examining the ultrastructural organization of the trypanosome endomembrane system. The application of functional data using inhibitors, such as latrunculin A for actin depolymerization, further strengthens the study by providing insights into the dynamics and regulatory mechanisms of the endomembrane system. This demonstrates how the actomyosin system contributes to cellular morphology and trafficking processes. Furthermore, the discovery of TbMyo1 localization to glycosomes introduces a novel aspect to the potential roles of myosin I proteins within the cell, particularly in the context of organelles analogous to peroxisomes. This observation not only broadens our understanding of myosin I functionality but also opens up new avenues for research into the cell biology of trypanosomatids, marking a significant contribution to the field.

A significant initial weakness was the reliance on spatial association data to infer functional relationships without direct demonstration of biochemical activities in vivo. The authors have since addressed this by including new evidence from TbMyo1 RNAi cell lines and EM data that show the effects of TbMyo1 depletion on cellular ultrastructure. The authors' responses and additional data reinforce their initial conclusions and address previous concerns. Several new, elegant hypotheses are proposed in the discussion that warrant further investigation to fully understand TbMyo1's interactions and regulatory mechanisms in vivo.

---

## [Referee Report · Reviewer #3 (Public Review)]

Summary:

In this work, Link and colleagues have investigated the localization and function of the actomyosin system in the parasite *Trypanosoma brucei*, which represents a highly divergent and streamlined version of this important cytoskeletal pathway. Using a variety of cutting-edge methods, the authors have shown that the *T. brucei* Myo1 homolog is a dynamic motor that can translocate actin, suggesting that it may not function as a more passive crosslinker. Using expansion microscopy, iEM, and CLEM, the authors show that MyoI localizes to the endosomal pathway, specifically the portion tasked with internalizing and targeting cargo for degradation, not the recycling endosomes. The glycosomes also appear to be associated with MyoI, which was previously not known. An actin chromobody was employed to determine the localization of filamentous actin in cells, which was correlated with the localization of Myo1. Interestingly, the pool of actomyosin was not always closely associated with the flagellar pocket region, suggesting that portions of the endolysomal system may remain at a distance from the sole site of parasite endocytosis. Lastly, the authors used actin-perturbing drugs to show that disrupting actin causes a collapse of the endosomal system in *T. brucei*, which they have shown recently does not comprise distinct compartments but instead a single continuous membrane system with subdomains containing distinct Rab markers.

Strengths:

Overall, the quality of the work is extremely high. It contains a wide variety of methods, including biochemistry, biophysics, and advanced microscopy that are all well deployed to answer the central question. The data is also well quantitated to provide additional rigor to the results. The main premise, that actomyosin is essential for the overall structure of the *T. brucei* endocytic system, is well supported and is of general interest, considering how uniquely configured this pathway is in this divergent eukaryote and how important it is to the elevated rates of endocytosis that are necessary for this parasite to inhabit its host.

Comments on revised version:

The revised manuscript has addressed the main issue, the lack of TbMyo1 functional data that was brought up during the first round of review. I find it interesting that Myo1 depletion has what appears to be a limited effect on endocytosis while producing a similar fragmentation of the endocytic pathway to what is seen with the LatA treatments. As CCV remains in both LatA treatments and TbMyo1 RNAi, it seems apparent that the organization of the endocytic pathway is not required for at least basal levels of endocytosis.

My other points were well addressed by the rebuttal. I am satisfied with the update.

---

## [Author Response]

The following is the authors’ response to the original reviews.

**eLife assessment**
This important study builds on a previous publication (with partially overlapping authors), demonstrating that *T. brucei* has a continuous endomembrane system, which probably facilitates high rates of endocytosis. Using a range of cutting-edge approaches, the authors present compelling evidence that an actomyosin system, with the myosin TbMyo1 as the molecular motor, is localized close to the endosomal system in the bloodstream form (BSF) of *Trypanosoma brucei*. It shows convincingly that actin is important for the organization and integrity of the endosomal system, and that the trypanosome Myo1is an active motor that interacts with actin and transiently associates with endosomes, but a role of Myo1 in endomembrane function in vivo was not directly demonstrated. This work should be of interest to cell biologists and microbiologists working on the cytoskeleton, and unicellular eukaryotes.

We were delighted at the editors’ positive assessment and the reviewers’ rigorous, courteous, and constructive responses to the paper. We agree that a direct functional role for TbMyo1 in endomembrane activity was not demonstrated in the original submission, but have incorporated some new data (see new supplemental Figure S5) using the TbMyo1 RNAi cell line which are consistent with our earlier observations and interpretations.

**Public Reviews:**

**Reviewer #1 (Public Review):**
Using a combination of cutting-edge high-resolution approaches (expansion microscopy, SIM, and CLEM) and biochemical approaches (in vitro translocation of actin filaments, cargo uptake assays, and drug treatment), the authors revisit previous results about TbMyo1 and TbACT in the bloodstream form (BSF) of *Trypanosoma brucei*. They show that a great part of the myosin motor is cytoplasmic but the fraction associated with organelles is in proximity to the endosomal system. In addition, they show that TbMyo1 can move actin filaments in vitro and visualize for the first time this actomyosin system using specific antibodies, a "classical" antibody for TbMyo1, and a chromobody for actin. Finally, using latrunculin A, which sequesters G-actin and prevents F-actin assembly, the authors show the delocalization and eventually the loss of the filamentous actin signal as well as the concomitant loss of the endosomal system integrity. However, they do not assess the localization of TbMyo1 in the same conditions.Overall the work is well conducted and convincing. The conclusions are not over-interpreted and are supported by the experimental results.

We are very grateful to Reviewer1 for their balanced assessment. The reviewer is correct that we did not assess the localisation of TbMyo1 following latrunculin A treatment, but it is worth noting that Spitznagel et al. carried out this exact experiment in the earlier 2010 paper – we have mentioned this in the revised manuscript.

**Reviewer #2 (Public Review):**
Summary:The study by Link et al. advances our understanding of the actomyosin system in *T. brucei*, focusing on the roleof TbMyo1, a class I myosin, within the parasite's endosomal system. Using a combination of biochemical fractionation, in vitro motility assays, and advanced imaging techniques such as correlative light and electron microscopy (CLEM), this paper demonstrates that TbMyo1 is dynamically distributed across early and late endosomes, the cytosol, is associated with the cytoskeleton, and a fraction has an unexpected association with glycosomes. Notably, the study shows that TbMyo1 can translocate actin filaments at velocities suggesting an active role in intracellular trafficking, potentially higher than those observed for similar myosins in other cell types. This work not only elucidates the spatial dynamics of TbMyo1 within *T. brucei* but also suggests its broader involvement in maintaining the complex architecture of the endosomal network, underscoring the critical role of the actomyosin system in a parasite that relies on high rates of endocytosis for immune evasion.Strengths:A key strength of the study is its exceptional rigor and successful integration of a wide array of sophisticated techniques, such as in vitro motility assays, and advanced imaging methods, including correlative light and electron microscopy (CLEM) and immuno-electron microscopy. This combination of approaches underscores the study's comprehensive approach to examining the ultrastructural organization of the trypanosome endomembrane system. The application of functional data using inhibitors, such as latrunculin A for actin depolymerization, further strengthens the study by providing insights into the dynamics and regulatory mechanisms of the endomembrane system. This demonstrates how the actomyosin system contributes to cellular morphology and trafficking processes. Furthermore, the discovery of TbMyo1 localization to glycosomes introduces a novel aspect to the potential roles of myosin I proteins within the cell, particularly in the context of organelles analogous to peroxisomes. This observation not only broadens our understanding of myosin I functionality but also opens up new avenues for research into the cellular biology of trypanosomatids, marking a significant contribution to the field.

We are very pleased that the Reviewer felt the work is a significant contribution to the state of the art.

Weaknesses:Certain limitations inherent in the study's design and scope render the narrative incomplete and make it challenging to reach definitive conclusions. One significant limitation is the reliance on spatial association data, such as colocalization of TbMyo1 with various cellular components-or the absence thereof-to infer functional relationships. Although these data suggest potential interactions, the authors do not confirm functional or direct physical interactions.While TbMyo1's localization is informative, the authors do not directly demonstrate its biochemical or mechanical activities in vivo, leaving its precise role in cellular processes speculative. Direct assays that manipulate TbMyo1 levels, activity, and/or function, coupled with observations of the outcomes on cellular processes, would provide more definitive evidence of the protein's specific roles in *T. brucei*. A multifaceted approach, including genetic manipulations, uptake assays, kinetic trafficking experiments, and imaging, would offer a more robust framework for understanding TbMyo1's roles. This comprehensive approach would elucidate not just the "what" and "where" of TbMyo1's function but also the "how" and "why," thereby deepening our mechanistic insights into *T. brucei's* biology.

The reviewer is absolutely correct that the study lacks data on direct or indirect interactions between TbMyo1 and its intracellular partners, and this is an obvious area for future investigation. Given the generally low affinities of motor-cargo interactions, a proximity labelling approach (such has already been successfully used in studies of other myosins) would probably be the best way to proceed.

The reviewer is also right to highlight that a detailed mechanistic understanding of TbMyo1 function in vivo is currently lacking. We feel that this would be beyond the scope of the present work, but have included some new data using the TbMyo1 RNAi cell line (Figure S5), which are consistent with our previous findings.

**Reviewer #3 (Public Review):**
Summary:In this work, Link and colleagues have investigated the localization and function of the actomyosin system in the parasite *Trypanosoma brucei*, which represents a highly divergent and streamlined version of this important cytoskeletal pathway. Using a variety of cutting-edge methods, the authors have shown that the *T. brucei* Myo1 homolog is a dynamic motor that can translocate actin, suggesting that it may not function as a more passive crosslinker. Using expansion microscopy, iEM, and CLEM, the authors show that MyoI localizes to the endosomal pathway, specifically the portion tasked with internalizing and targeting cargo for degradation, not the recycling endosomes. The glycosomes also appear to be associated with MyoI, which was previously not known. An actin chromobody was employed to determine the localization of filamentous actin in cells, which was correlated with the localization of Myo1. Interestingly, the pool of actomyosin was not always closely associated with the flagellar pocket region, suggesting that portions of the endolysomal system may remain at a distance from the sole site of parasite endocytosis. Lastly, the authors used actin-perturbing drugs to show that disrupting actin causes a collapse of the endosomal system in *T. brucei*, which they have shown recently does not comprise distinct compartments but instead a single continuous membrane system with subdomains containing distinct Rab markers.Strengths:Overall, the quality of the work is extremely high. It contains a wide variety of methods, including biochemistry, biophysics, and advanced microscopy that are all well-deployed to answer the central question. The data is also well-quantitated to provide additional rigor to the results. The main premise, that actomyosin is essential for the overall structure of the *T. brucei* endocytic system, is well supported and is of general interest, considering how uniquely configured this pathway is in this divergent eukaryote and how important it is to the elevated rates of endocytosis that are necessary for this parasite to inhabit its host.

We are very pleased that the Reviewer formed such a positive impression of the work.

Weaknesses:(1) Did the authors observe any negative effects on parasite growth or phenotypes like BigEye upon expression of the actin chromobody?

Excellent question! There did appear to be detrimental effects on cell morphology in some cells, and it would definitely be worth doing a time course of induction to determine how quickly chromobody levels reach their maximum. The overnight inductions used here are almost certainly excessive, and shorter induction times would be expected to minimise any detrimental effects. We have noted these points in the Discussion.

(2) The Garcia-Salcedo EMBO paper cited included the production of anti-actin polyclonal antibodies that appeared to work quite well. The localization pattern produced by the anti-actin polyclonals looks similar to the chromobody, with perhaps a slightly larger labeling profile that could be due to differences in imaging conditions. I feel that the anti-actin antibody labeling should be expressly mentioned in this manuscript, and perhaps could reflect differences in the F-actin vs total actin pool within cells.

Implemented. We have explicitly mentioned the use of the anti-actin antibody in the Garcia-Salcedo paper in the revised Results and Discussion sections.

(3) The authors showed that disruption of F-actin with LatA leads to disruption of the endomembrane system, which suggests that the unique configuration of this compartment in *T. brucei* relies on actin dynamics. What happens under conditions where endocytosis and endocyctic traffic is blocked, such as 4 C? Are there changes to the localization of the actomyosin components?

Another excellent question! We did not analyse the localisation of TbMyo1 and actin under temperature block conditions, but this would definitely be a key experiment to do in follow-up work.

(4) Along these lines, the authors suggest that their LatA treatments were able to disrupt the endosomal pathway without disrupting clathrin-mediated endocytosis at the flagellar pocket. Do they believe that actin is dispensable in this process? That seems like an important point that should be stated clearly or put in greater context.

Whether actin plays a direct or indirect role in endocytosis would be another fascinating question for future enquiry, and we do not have the data to do more than speculate on this point. Recent work in mammalian cells (Jin et al., 2022) has suggested that actin is primarily recruited when endocytosis stalls, and it could be that a similar role is at play here. We have noted this point in the Discussion. The observation of clathrin vesicles close to the flagellar pocket membrane and clathrin patches on the flagellar pocket membrane itself in the LatA-treated cells might suggest that some endocytic activity can occur in the absence of filamentous actin.

**Recommendations for the authors:**

**Note from the Reviewing Editor:**
During discussion, all reviewers agreed that the role of TbMyo1 in vivo in endomembrane function had not been directly demonstrated. This could be done by testing the endocytic trafficking of (for example) fluorophoreconjugated TfR and BSA in the existing Myo1 RNAi line, using wide-field microscopy. Examining the endosomes/lysosomes' organization by thin-section EM would be even better. The actin signal detected by the chromobody tends to occupy a larger region than the MyoI. It's therefore conceivable that actin filamentation and stabilization via other actin-interacting proteins create the continuous endosomal structure, while MyoI is necessary for transport or other related processes.

These are all excellent points and very good suggestions. We have now incorporated new data (supplemental Figure S5) that includes BSA uptake assays in the TbMyo1 RNAi cell line and electron microscopy imaging after TbMyo1 depletion – the results are consistent with our earlier observations.

**Reviewer #1 (Recommendations For The Authors):**
- Figure S2E. This panel is supposed to show the downregulation of TbMyo1 in the PCF compared to BSF but there is no loading control to support this claim. This is important because the authors mention in lines 381-383 that this finding conflicts with the previous study (Spitznagel et al., 2010). The authors also indicate in the figure legend that there is 50% less signal but there is no explanation about this quantification.

Good point. Equal numbers of cells were loaded in each lane, but we did not have an antibody against a protein known to be expressed at the same level in both PCF and BSF cells to use as a loading control. Using a total protein stain would have been similarly unhelpful in this context, as the proteomes of PCF and BSF cells are dissimilar. The quantification was made by direct measurement after background subtraction, but without normalisation owing to the lack of a loading control. This makes the conclusion somewhat tentative, but given the large difference in signal observed between the two samples (and the fact that this is consistent with the proteomic data obtained by Tinti and Ferguson) we feel that the conclusion is valid. We have clarified these points in the figure legend and Discussion.

- It is mentioned in the discussion, as unpublished observations, that the predicted FYVE motif of TbMyo1 can bind specifically PI(3)P lipids. This is a very interesting point that would be new and would strengthen the suggested association with the endosomal system mainly based on imaging data.

We agree that this is – potentially – a very exciting observation and it is an obvious direction for future enquiry.

The data are preliminary at this stage and will form the basis of a future publication. Given that the predicted FYVE domain of TbMyo1 and known lipid-binding activity of other class I myosins makes this activity not wholly unexpected, we feel that it is acceptable at this stage to highlight these preliminary findings.

- The authors use the correlation coefficient to estimate the colocalization (lines 223-226). Although they clearly explain the difference between the correlation coefficient and the co-occurrence of two signals, I wonder if it would not be clearer for the audience to have quantification of the overlapping signals. Also, it is not mentioned on which images the correlation coefficient was measured. It seems that it is from widefield images (Figures 3E and 6E), and likely from SIM images for Figure 3C but the resolution is different. Are widefield images sufficient to assess these measurements?

With hindsight, and given the different topological locations of TbMyo1 and the cargo proteins (cytosolic and lumenal, respectively) it would probably have been wiser to measure co-occurrence rather than correlation, but we would prefer not to repeat the entire analysis at this stage. The correlations were measured from widefield images using the procedure described in the Materials & Methods. These are obviously lower resolution than confocal or SIM images would be, but are still of value, we believe. One further point – upon re-examination of some of the TbMyo1 transferrin (Tf) and BSA data, we noticed that there are many pixels with a value of 0 for Tf/BSA and a nonzero value for TbMyo1 and vice-versa. The incidence of zero-versus-nonzero values in the two channels will have lowered the correlation coefficient, and in this sense, the correlation coefficients are giving us a hint of what the immuno-EM images later confirm: that the TbMyo1 and cargo are present in the same locations, but in different proportions. We have added this point to the discussion.

- It would be good to know if the loss of the endosomal system integrity (using EBI) is the same upon TbMyo1 depletion than in the latrunculin A treated parasites.

We agree! We have now included new data (Figure S5) that suggests endosomal system morphology is altered upon TbMyo1 depletion. We would predict that the effect upon TbMyo1 depletion is slower or less dramatic than upon LatA treatment (as LatA affects both actin and TbMyo1, given that TbMyo1 depends upon actin for its localisation).

- Conversely, it would be of interest to see how the localization of TbMyo1 changes upon latrunculin A treatment.

This experiment was done in 2010 by Spitznagel et al., who observed a delocalisation of the TbMyo1 signal after LatA treatment. We have noted this in the Results and Discussion.

Minor corrections:- Line 374: Figure S1 should be Figure S2.

Implemented (many thanks!).

- Panel E of Figure S2 refers to TbMyo1 and should therefore be included in Figure S1 and not S2.

We would prefer not to implement this suggestion. We did struggle over the placing of this panel for exactly this reason, but as the samples were obtained as part of the experiments described in Figure S2, we felt that its placement here worked best in terms of the narrative of the manuscript.

- Figure S2F: the population of TbMyo21 +Tet seems lost after 48 h although the authors mention that there is no growth defect.

Good eyes! We have re-added the panel, which shows that there was no growth defect in the tetracycline-treated population.

**Reviewer #2 (Recommendations For The Authors):**
Fig 1 vs. Figure 3: The biochemical fractionation experiments have been well-controlled, showing that 40% of TbMyo1 is found in both the cytosolic and cytoskeletal fractions, with only 20% in the organelle-associated fraction. The conclusion is supported by the experimental design, which includes controls to rule out crosscontamination between fractions. However, does this contrast with the widefield microscopy experiments, where the vast majority of the signal is in endocytic compartments and nowhere else?

This is a good point. There are three factors that probably explain this. First, given that the actin cytoskeleton is associated with the endosomal system, a large proportion of the material partitioning into the cytoskeleton (P2) fraction is probably localised to the endosomal system (a fun experiment would be to repeat the fractionation with addition of ATP to the extraction buffer to make the myosin dissociate and see whether more appeared in the SN2 fraction as a result). Second, the 40% of the TbMyo1 that is cytosolic is distributed throughout the entire cellular volume, whereas the material localised to the endosomes is concentrated in a much smaller space, by comparison, and producing a stronger signal. Third, the widefield microscopy images have had brightness and contrast adjusted in order to reduce “background” signal, though this will also include cytosolic molecules. We hope these explanations are satisfactory, but would welcome any additional thoughts from either the reviewer or the community.

The section title 'TbMyo1 translocates filamentous actin at 130 nm/s' could mislead readers by not specifying that the findings are from an in vitro experiment with a recombinant protein, which may not fully reflect the cell's complex context. Although this detail is noted in the figure legend, incorporating it into the main text and considering a title revision would ensure clarity and accuracy.

Good point. Implemented – we have amended the section title to “TbMyo1 translocates filamentous actin at 130 nm/s in vitro” and the figure legend title to “TbMyo1 translocates filamentous actin in vitro”.

The discussion of the translocation experiment could be better phrased addressing certain limitations. The in vitro conditions might not fully capture the complexity and dynamic nature of cellular environments where multiple regulatory mechanisms, interacting partners, and cellular compartments come into play.

Good point, implemented. We have added a note on this to the Discussion.

It is puzzling that RNAi, which is widely used in *T. brucei* was not used to further investigate the functional roles of TbMyo1 in *Trypanosoma brucei*. Given that the authors already had the cell line and used it to validate the specificity of the anti-TbMyo1. RNAi could have been employed to knock down TbMyo1 expression and observe the resultant effects on actin filament dynamics and organization within the cell. This would have directly tested TbMyo1's contribution to actin translocation observed in the in vitro experiments.

It would obviously be interesting to carry out an in-depth characterisation of the phenotype following TbMyo1 depletion and whether this has an effect on actin dynamics. We have now included additional data (supplemental Figure S5) using the TbMyo1 RNAi cells and the results are consistent with our earlier observations and interpretations. It is worth noting too that at least for electron microscopy studies of intracellular morphology, the slower onset of an RNAi phenotype and the asynchronous replication of *T. brucei* populations make observation of direct (early) effects of depletion challenging – hence the preferential use of LatA here to depolymerise actin and trigger a faster phenotype.

I found that several declarative statements within the main text may not be fully supported by the overall evidence. I suggest modifications to present a more balanced view,Line 227: "The results here suggest that although the TbMyo1 distribution overlaps with that of endocytic cargo, the signals are not strongly correlated." This conclusion about the lack of strong correlation might mislead readers about the functional relationship between TbMyo1 and endocytic cargo, as colocalization does not directly imply functional interaction.

We would prefer not to alter this statement. It was our intention to phrase this cautiously, as we have not directly investigated the functional interplay between TbMyo1 and endocytic cargo and the subsequent sentence directs the reader to the Discussion for more consideration of this issue.

Line 397: "This relatively high velocity might indicate that TbMyo1 is participating in intracellular trafficking of BSF *T. brucei* and functioning as an active motor rather than a static tether." The statement directly infers TbMyo1's functional role from in vitro motility assay velocities without in vivo corroboration.

We have amended the sentence in the Discussion to make it clear that it is speculative.

The hypothesis that cytosolic TbMyo1 adopts an auto-inhibited "foldback" configuration, drawn by analogy with findings from other studies, is intriguing. Yet, direct evidence linking this configuration to TbMyo1's function in *T. brucei* is absent from the data presented.

We have amended the sentence in the Discussion to make it clear that it is speculative. Future in vitro experiments will test this hypothesis directly.

The suggestion that a large cytosolic fraction of TbMyo1 indicates dynamic behavior, high turnover on organelles, and a low duty ratio is plausible but remains speculative without direct experimental evidence. Measurements of TbMyo1 turnover rates or duty ratios in *T. brucei* through kinetic studies would substantiate this claim with the necessary evidence.

We have amended the sentence in the Discussion to make it clear that it is speculative, and deleted the reference to a possible low duty ratio. Again, future in vitro experiments will measure the duty ratio of TbMyo1 using stopped-flow.

**Reviewer #3 (Recommendations For The Authors):**
Lines 171-172: The authors mention that MyoI could be functioning as a motor rather than a tether. The differences in myosin function have not been introduced prior to this. I would recommend explaining these differences and what it could mean for the function of the motor in the introduction to help a non-expert audience.

Good point. Implemented.

Line 94-95: This phenotype only holds for the bloodstream form- the procyclic form are quite resistant to actin RNAi and MyoI RNAi. I would clarify.

Good point. Implemented.

Line 142-146: did the authors attempt to knock out the Myo21?

Good point. No, this was not attempted. Given the extremely low expression levels of TbMyo21 in the BSF cells we would not expect a strong phenotype, but this assumption would be worth testing.

Figure 3D: is there a reason why the authors chose to show the single-channel images in monochrome in this case?

Not especially. These panels are the only ones that show a significant overlap in the signals between the two channels (unlike the colabelling experiments with ER, Golgi), so greyscale images were used because of their higher contrast.

Line 397-398: I'm struggling a bit to understand how MyoI could be involved in intracellular trafficking in the endosomal compartments if the idea is that we have a continuous membrane? Some more detail as to the author's thinking here would be useful.

Implemented. We have noted that this statement is speculative, and emphasised that being an active motor does not automatically mean that it is involved in intracellular traffic – it could instead be involved in manipulating endosomal membranes. We have noted too that the close proximity between TbMyo1 and the lysosome (Figures 3-5) could be important in this regard. The lysosome is not contiguous with the endosomal system, and it is possible that TbMyo1 is working as a motor to transport material (class II clathrin-coated vesicles) from the endosomal system to the lysosome.

Line 493-496: Does this mean that endocytosis from the FP does not require actin? This would be hard to explain considering the phenotypes observed in the original actin RNAi work. Is the BigEye phentopye observed in BSF actin RNAi and Myo1 RNAi cells due to some indirect effect?

It seems possible that actin is not directly or essentially involved in endocytosis, and the characterisation of the actin RNAi phenotype would be worth revisiting in this respect – we have noted this in the Discussion. Although RNAi of actin was lethal, the phenotype appears less penetrant than that seen following depletion of the essential endocytic cofactor clathrin (based on the descriptions in Garcia-Salcedo et al., 2004 and Allen et al., 2003). BigEye phenotypes occur in BSF cells whenever there is some perturbation of endomembrane trafficking and are not necessarily a direct consequence of depletion – this is why careful investigation of early timepoints following RNAi induction is critical.